# VBA: Vector Bundle Attention for Intrinsically Geometry-Aware Learning

## Abstract

Learning from geometrically structured data is fundamental in biology, physics, and computer vision. Graph Neural Networks capture local structure but are limited by message passing, while Transformers model global dependencies yet often neglect geometry. We introduce the Vector Bundle Attention Transformer (VBA-Transformer), a framework that redefines attention as an intrinsic geometric operator. Each token couples a base manifold coordinate with a fiber feature vector, following vector bundle theory. A principled parallel transport mechanism aligns fiber features before similarity is computed, embedding geometry directly into the attention operation. Unlike prior models that inject geometry as an external bias, VBA integrates it natively. On challenging single-cell RNA sequencing benchmarks, our model achieves state-of-the-art accuracy, with significant gains over Transformer-based baselines (over 3%-5%). On spatial transcriptomics, it demonstrates superior clustering performance. On 3D point clouds, VBA achieves competitive accuracy, validating its broad generalization capabilities. Beyond empirical gains, VBA provides theoretical analyses (invariance of the ideal operator and perturbation bounds), and empirical evidence of stability for the practical implementation, establishing geometric disentanglement as a powerful new principle for a versatile architecture poised for broad impact.

## 1 Introduction

Learning representations from non-Euclidean geometric data is a central challenge in machine learning (Bronstein et al., 2017; 2021). Such data is ubiquitous, ranging from molecular graphs and 3D point clouds to social networks and biological tissues. However, existing paradigms often struggle to capture these intrinsic geometries effectively. While Graph Neural Networks (GNNs) and Transformers represent the dominant paradigms for learning on structured data, both are constrained by inherent limitations. GNNs (Scarselli et al., 2008; Defferrard et al., 2016; Velickovic et al., 2017) are inherently geometric, operating through a message passing over a predefined graph. However, this reliance on a fixed, often heuristically defined adjacency matrix makes them structurally rigid and limits their ability to capture complex, long-range dependencies or to learn the underlying manifold itself (Xu et al., 2018; Maron et al., 2018; Zhou et al., 2020; Zheng et al., 2024). In contrast, Transformers (Vaswani et al., 2017) have demonstrated unparalleled success in modeling complex dependencies in sequential and set data, but they are fundamentally "geometry-blind," assuming a fully-connected graph and lacking an intrinsic understanding of locality or geometric relationships(Ying et al., 2021; Yuan et al., 2025).

Recent studies have attempted to bridge this gap by injecting geometric information as additive biases in attention (Ying et al., 2021; Chen et al., 2022; 2024; Yang et al., 2024). While effective, these approaches represent only a weak integration of content and geometry (Yuan et al., 2025). The core dot-product attention still operates in the ambient feature space, with geometry acting as an external correction. This raises a central question: **Can we redefine attention itself as an intrinsic geometric operator within a learned manifold?**

We answer this challenge with the **Vector Bundle Attention Transformer (VBA-Transformer)**. Our core insight is that a truly powerful geometric model should not passively accept a predefined structure, but should instead simultaneously learn the underlying manifold of the data and a metric for measuring affinity within that manifold. To achieve this, we turn to the mathematical framework

of vector bundles from differential geometry(Nakahara, 2018; Bamberger et al., 2024; Liu & Su, 2015). This allows us to rigorously model data points as residing within a "vector bundle structure," which inherently combines a low-dimensional base manifold (capturing global spatial arrangement) with a high-dimensional fiber space at each point (encoding its intricate local state). Crucially, within this learned vector bundle, we introduce a novel attention mechanism in which Query and Key vectors are formed from feature projections and aligned via parallel transport induced by the learned geometric coordinates. Consequently, the attention score is no longer a measure of feature similarity in an arbitrary ambient space, but a learnable metric of affinity within the learned vector bundle space itself. This fundamentally redefines the attention mechanism as an intrinsic geometric operator, tightly coupling representation learning with manifold discovery.

We instantiate our approach as an end-to-end autoencoder and validate the VBA-Transformer on diverse benchmarks. Specifically, we focus on biological data with complex intrinsic geometries: spatial transcriptomics (ST) datasets, which capture gene expression while preserving spatial location (Heydari & Sindi, 2023), and single-cell RNA sequencing (scRNA-seq) data, where cells form an implicit manifold in high-dimensional gene expression space (Moon et al., 2018; Xu et al., 2023). Beyond biological domains, we validate VBA's universality through evaluation on ModelNet40, the canonical 3D point cloud classification task.

Our contributions are summarized as follows:

1. **Vector Bundle Attention (VBA):** We introduce a theoretically grounded attention mechanism that operates intrinsically on a learned geometric manifold.

2. **VBA-Transformer:** We present a new architecture that effectively disentangles geometry from features, endowing Transformers with native geometric awareness.

3. **Theory:** We establish invariance properties and derive perturbation bounds, providing stability guarantees and a principled foundation for geometric learning.

4. **Empirical Validation:** We achieve state-of-the-art performance on scRNA-seq and highly competitive results on spatial transcriptomics, and demonstrate broad generality on 3D point-cloud classification.

## 2 RELATED WORKS

Our work is positioned at the intersection of Graph Neural Networks, Transformers for geometric data, and the broader field of Geometric Deep Learning.

**Graph Neural Networks (GNNs).** Graph Neural Networks (GNNs) form the classical approach for learning on structured data (Scarselli et al., 2008; Defferrard et al., 2016; Kipf & Welling, 2017; Velickovic et al., 2017). And most GNNs fall into Message Passing Neural Networks (MPNNs) (Gilmer et al., 2017), where nodes exchange messages with their immediate neighbors and update their representations by aggregating these local features. While inherently geometric, they are tightly coupled to a predefined adjacency structure, which often limits their ability to capture long-range dependencies or recover the underlying manifold (Xu et al., 2018; Zhou et al., 2020) and leading to issues like over-smoothing and limited expressivity (Li et al., 2018; Oono & Suzuki, 2019; Xu et al., 2018; Morris et al., 2019). To address this issue, some research has been conducted into graph structure learning, where the adjacency is adaptively inferred from data(Zheng et al., 2024; Zhao et al., 2023; Franceschi et al., 2019). While these methods improve flexibility, they remain within the message-passing paradigm. This makes them less suitable for disentangling geometry from content.

**Transformers on Graphs and Geometric Data.** Transformers (Vaswani et al., 2017) have recently been adapted to structured domains, thanks to their ability to model long-range interactions. Early works such as Graph-BERT treated graphs as sequences, losing much of the structural information (Zhang et al., 2020). Later models like Graphormer, GPS and GBT(Ying et al., 2021; Rampášek et al., 2022; Venkat et al., 2023) incorporate centrality and spatial encodings into the attention mechanism, while approaches for 3D point clouds(Guo et al., 2021; Chen et al., 2025) directly encode geometric relationships as attention biases. Although these designs enrich attention with geometric priors, they still treat geometry as an auxiliary signal added onto a standard

Transformer. However, in all these cases, geometry acts as an external correction, while the core dot-product attention still operates in the ambient feature space, rendering these approaches fundamentally "geometry-weak".

**Geometric Deep Learning.** A parallel direction seeks to endow models with stronger geometric foundations. Hyperbolic and manifold-based neural networks (Nickel & Kiela, 2017; Chami et al., 2019; Sala et al., 2018) redefine representation learning in curved spaces, while bundle-based methods propose to model data using fiber structures over a base manifold (Bamberger et al., 2024). These works demonstrate the promise of non-Euclidean approaches, yet they either lack the scalability of Transformer architectures or fail to integrate geometry directly into the attention mechanism.

**Our Position.** Unlike prior approaches that introduce geometry through external positional encodings or apply geometric corrections after attention, VBA-Transformer embeds geometry inside the attention mechanism itself. Each token is represented as a point on a learned base manifold with an attached fiber vector, and key/value features are first aligned by an isometric transport before similarity is computed. This "transport-then-attend" formulation differs from both positional-bias Transformers and gauge/message-passing models, and enables principled manifold discovery while preserving the expressive power and scalability of Transformers.

## 3 METHOD

We propose **Vector Bundle Attention (VBA)**, a Transformer attention operator that is defined *intrinsically* on a learned vector bundle over a learned base manifold. Rather than treating geometry as an auxiliary bias, VBA rewrites the elementary attention operations (projection, similarity, aggregation) to operate on fiber-valued features and uses a learnable *connection* (parallel transport) to align fiber vectors before similarity computation. This yields an attention operator that couples *content* and *geometry* at the operator level, enabling the model to learn non-flat geometries (via a curvature proxy) and to trade off efficiency vs expressivity via bundle-mixing and low-rank/residual parameterizations. The architecture of the VBA is shown in Figure 1

.

### 3.1 VECTOR BUNDLE ATTENTION

Given an input sequence of ambient features $X = \{x_i\}_{i=1}^N$ where $x_i \in \mathbb{R}^D$, a VBA layer performs the following conceptual steps, which are detailed below and summarized in Algorithm 1.

1. **Projection:** Each input $x_i$ is projected into a disentangled representation consisting of a coordinate on a latent base manifold, $b_i \in \mathbb{R}^{d_b}$, and a feature vector in a local fiber space, $f_i \in \mathbb{R}^{d_f}$.

2. **Curvature Correction:** The fiber vector is optionally modulated by a learnable curvature term, allowing the model to capture non-flat geometries.

3. **Parallel Transport:** For any pair of points $(i, j)$, a learnable connection defines a parallel transport operator $T_{j \to i}$ that moves the fiber vector at point $j$ to the fiber space at point $i$.

4. **Fiber Attention:** Attention is then computed in the fiber space, using the transported vectors to calculate similarity and aggregate values.

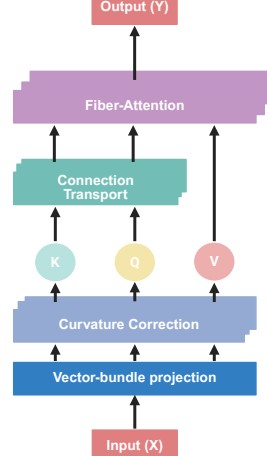

Figure 1: Overview of Vector Bundle Attention (VBA).

**Projection to the Vector Bundle.** We first realize the vector bundle decomposition via linear projections. Each input token $x_i$ is mapped to a base coordinate $b_i$ and a set of $M$ candidate fiber vectors:

$$b_i = W_b x_i \in \mathbb{R}^{d_b}, \qquad f_i^{(m)} = W_f^{(m)} x_i \in \mathbb{R}^{d_f} \quad (m = 1, \ldots, M). \tag{1}$$

To allow for flexible representations, we use a data-dependent gating mechanism to dynamically combine these candidate fibers. A small bundle-selector network $s : \mathbb{R}^D \to \mathbb{R}^M$ produces mixing weights $\alpha_i = \text{softmax}(s(x_i))$. These are used to produce the final fiber representation $\bar{f}_i$, which is passed to the subsequent stages:

$$\bar{f}_i = \text{LayerNorm}\left(\sigma(g(x_i)) \odot \phi\left(\sum_{m=1}^{M} \alpha_i^{(m)} A^{(m)} f_i^{(m)}\right)\right), \tag{2}$$

where $g(\cdot)$ is a linear projection, $\sigma$ is a sigmoid gate, $\phi$ is a GELU non-linearity. Crucially, the matrices $\{A^{(m)}\}_{m=1}^{M}$ function as learnable, bundle-specific feature transformations. This multi-bundle design allows the model to process input features along $M$ parallel pathways, learning specialized refinements for different aspects of the data before they are adaptively combined. This design choice ensures that the fiber refinement is not an arbitrary feature transformation, but a geometrically-informed process that is directly coupled to the parameters of the underlying vector bundle structure.

**Curvature Correction.** To provide the model with the most principled geometric information, we compute a high-fidelity approximation of the formal curvature 2-form, $\Omega = d\Gamma + \Gamma \wedge \Gamma$. This is made possible by our learnable connection field, implemented as a neural network `ConnectionNet` which maps any base coordinate $b$ to the connection coefficient matrices $\{\Gamma_k(b)\}$.

The two components of the curvature tensor are computed as follows:

- The derivative term, $d\Gamma$, is calculated by taking the Jacobian of the `ConnectionNet`'s output with respect to the input coordinates $b$, which is computed efficiently via automatic differentiation.
- The algebraic term, $\Gamma \wedge \Gamma$, is calculated using the commutator (Lie bracket) of the output connection matrices $[\Gamma_1(b), \Gamma_2(b)]$.

These two terms are then combined to form a position-dependent curvature tensor $\Omega(b)$. This tensor provides a rich, local description of the learned manifold's geometry, which can then be used to modulate the fiber representations. This approach moves beyond simple proxies and directly integrates a core component of differential geometry into the network's architecture.

**The Learnable Connection and Parallel Transport.** A central innovation of our work is an endpoint-conditioned isometric transport mechanism which is orthogonal, length-preserving, serving as a practical surrogate for formal path-dependent parallel transport. We design a `TransportNet` module that learns an orthogonal transport operator $T_{j \to i} \in SO(d_f)$ for each pair of points. This enforces isometry, ensuring that the transport operation is a pure rotation that preserves the length of the feature vectors.

To achieve this, the `TransportNet` uses a lightweight MLP to predict a generator matrix $S \in \mathbb{R}^{d_f \times d_f}$ from the base coordinates $(b_i, b_j)$. This matrix is then forced to be skew-symmetric:

$$S_{\text{skew}} = \frac{1}{2}(S - S^\top). \tag{3}$$

The final transport operator is the matrix exponential of this skew-symmetric matrix, which is guaranteed by construction to be a special orthogonal matrix (a rotation):

$$T_{j \to i} = \exp(S_{\text{skew}}). \tag{4}$$

This formulation allows the model to learn a complex, endpoint-dependent, and geometrically principled transport mechanism directly from data.

**Fiber Attention.** With the transport operator defined, we first project the (optionally curvature-corrected) fiber vectors $\bar{f}_i$ into queries, keys, and values: $Q_i = W_q \bar{f}_i$, $K_j = W_k \bar{f}_j$, $V_j = W_v \bar{f}_j$. To compute the attention score from query $i$ to key $j$, we first transport $K_j$ and $V_j$ to the fiber space at $i$:

$$\tilde{K}_{j \to i} = T_{j \to i} K_j, \qquad \tilde{V}_{j \to i} = T_{j \to i} V_j. \tag{5}$$

The attention logits and weights are then computed using the transported key:

$$e_{ij} = \frac{\langle Q_i, \tilde{K}_{j \to i} \rangle}{\sqrt{d_f}}, \qquad \alpha_{ij} = \text{softmax}_j(e_{ij}). \tag{6}$$

The final output fiber is a weighted sum of the transported value vectors:

$$y_i^{\text{fiber}} = \sum_j \alpha_{ij} \, \tilde{V}_{j \to i}. \tag{7}$$

This output is then projected back to the ambient space $\mathbb{R}^D$ with a final linear layer $W_o$: $y_i = x_i + W_o y_i^{\text{fiber}}$

## 3.2 MODEL ARCHITECTURE AND IMPLEMENTATION

The VBA block is designed as a drop-in replacement for standard self-attention within a Pre-LayerNorm (Pre-LN) Transformer architecture:

$$X' = X + \text{VBA}(\text{LN}(X)), \quad X'' = X' + \text{FFN}(\text{LN}(X')). \tag{8}$$

This modular design is made practical, stable, and efficient through several key choices:

- **Stable Initialization:** The transport operator $T$ is initialized near the identity matrix to ensure stable training dynamics from the start.
- **Expressivity:** Soft bundle mixing and data-dependent gating are employed to efficiently enhance the model's representational power.
- **Efficiency:** For large-scale inputs, computational cost is managed via low-rank parameterizations of the transport matrices or by restricting attention to local windows.

These features ensure that VBA is a robust operator that embeds a powerful geometric inductive bias at the core of the Transformer architecture.

# 4 EXPERIMENT

In this work, our proposed VBA-Transformer is designed to capture complex intrinsic geometric structures by disentangling geometry from content, making it especially suitable for data characterized by rich and often non-Euclidean relationships. We test this hypothesis by benchmarking our model against state-of-the-art baselines across three distinct domains: 1. Spatial Transcriptomics (ST), where data possesses an explicit spatial, near-Euclidean geometry. 2. Single-Cell RNA Sequencing (scRNA-seq), where the geometric relationships are implicit and must be learned from a high-dimensional manifold. 3. 3D Point Cloud Classification, a canonical benchmark for geometric deep learning that tests generalization to non-biological, explicit 3D structures.

## 4.1 COMMON EXPERIMENTAL SETTINGS.

We implement the VBA-Transformer using PyTorch. All experiments are conducted on four NVIDIA GH200 GPUs. For optimization, we use the AdamW optimizer with a learning rate of $10^{-4}$. A cosine annealing learning rate schedule with 10% warm-up epochs is applied. Training is performed for 200 epochs with a batch size of 64. A fixed random seed of 42 is used across all experiments for reproducibility. We evaluate performance using standard metrics relevant to each task, detailed in the respective subsections.

## 4.2 EXPERIMENTS ON SPATIAL TRANSCRIPTOMICS

The explicit geometric structure of spatial transcriptomics (ST) data presents a unique challenge and opportunity for representation learning (Heydari & Sindi, 2023). We first apply our model to identify spatial domains, a fundamental unsupervised learning task in the field. Success in this task demonstrates a model's ability to effectively integrate gene expression data with spatial information to uncover meaningful biological patterns in tissue.

**Data.** We evaluate the performance of our model on the widely-used human dorsolateral prefrontal cortex (DLPFC) dataset from the Lieber Institute for Brain Development (LIBD) (Maynard et al., 2021). This benchmark dataset contains 12 spatially-resolved transcriptomics slices, each

Table 1: Adjusted Rand Index (ARI) on 12 human DLPFC samples. The best performance per sample (row) is shown in **bold**. Our method, VBA-ST, demonstrates highly competitive performance.

| Sample | BayesSpace | SpaGCN | DeepST | GraphST | BASS | DiffusionST | VBA-ST |
|--------|-----------|--------|--------|---------|------|-------------|--------|
| 151507 | 0.47 | 0.46 | 0.50 | **0.51** | 0.50 | 0.43 | **0.51** |
| 151508 | 0.44 | 0.43 | 0.39 | 0.35 | 0.46 | 0.46 | **0.47** |
| 151509 | 0.40 | 0.47 | 0.36 | 0.41 | 0.51 | 0.54 | **0.61** |
| 151510 | 0.38 | 0.46 | 0.38 | 0.41 | 0.50 | **0.51** | 0.50 |
| 151669 | **0.47** | 0.37 | 0.35 | 0.27 | 0.35 | 0.38 | 0.42 |
| 151670 | 0.43 | 0.38 | 0.34 | 0.25 | 0.38 | **0.46** | 0.41 |
| 151671 | 0.49 | 0.56 | 0.52 | 0.50 | 0.56 | 0.58 | **0.65** |
| 151672 | 0.44 | 0.57 | 0.50 | **0.58** | 0.56 | 0.55 | 0.57 |
| 151673 | 0.55 | 0.52 | **0.55** | 0.51 | 0.53 | 0.54 | 0.53 |
| 151674 | 0.32 | 0.39 | 0.49 | 0.48 | 0.51 | **0.52** | 0.46 |
| 151675 | 0.53 | 0.47 | **0.57** | 0.52 | 0.55 | 0.46 | 0.47 |
| 151676 | 0.37 | 0.33 | 0.52 | 0.42 | **0.53** | 0.51 | 0.37 |

with expert-annotated ground-truth labels corresponding to the four or six distinct cortical layers and white matter (WM). Its well-organized, multi-layered tissue architecture provides an ideal testbed for evaluating a model's capacity to learn representations that preserve spatially coherent biological structures. In line with standard practice, we log-normalize the raw gene expression counts and use the top 3,000 highly variable genes (HVGs) as input features.

**Experimental Settings.** The task is to perform unsupervised spatial clustering to recover the annotated anatomical layers. We benchmark our VBA-Transformer model for spatial transcriptomics (VBA-ST) against six state-of-the-art baselines: BayesSpace(Zhao et al., 2021a), SpaGCN(Hu et al., 2021), DeepST(Xu et al., 2022), GraphST(Long et al., 2023), BASS(Li & Zhou, 2022), and DiffusionST(Cui et al., 2025). Baselines were executed using their officially suggested hyperparameters for a fair comparison. We quantify clustering accuracy using three standard metrics: the Adjusted Rand Index (ARI) is reported in the main paper as the primary performance metric; for a more comprehensive analysis, Normalized Mutual Information (NMI) and Completeness scores are provided in the Appendix B.2.

**Result.** As summarized in Table 1, across the 12 DLPFC samples, VBA-ST achieves the best ARI on 4 samples (151507, 151508, 151509, 151671) and ranks within the top-3 on 8 out of 12 samples. Notably, on samples 151509 and 151671, VBA-ST shows a substantial margin over the next best model (+0.07 in ARI), highlighting its strong ability to capture spatially coherent biological structures. Compared with Graph Convolutional Network (GCN)-based approaches such as SpaGCN and GraphST, which rely on iterative local aggregation and can be prone to over-smoothing, VBA-ST directly models global context and the intricate geometric relationships between all spots. This design enables it to learn more expressive representations, yielding state-of-the-art results on multiple samples and consistently competitive performance across diverse tissue sections. On average, VBA-ST (0.4983) performs better than the strongest baselines (DiffusionST and BASS 0.495), while also achieving the highest ARI on multiple samples.

## 4.3 EXPERIMENTS ON SPATIAL SCRNA-SEQ

Our next set of experiments aims to explore VBA-Transformer's generalization capabilities in a biological context where the geometric structure is more abstract and latent. To this end, we focus on the task of cell type annotation in single-cell RNA sequencing (scRNA-seq). It is widely hypothesized that distinct cell types form a complex, low-dimensional manifold within the high-dimensional transcriptomic space (Moon et al., 2018). Accurate cell type annotation is therefore not merely a classification problem, but fundamentally a task of geometric structure discovery. This is precisely the challenge VBA-Transformer is designed to address. Through its vector bundle attention mechanism, our model explicitly learns an underlying base manifold to capture the intrinsic, non-Euclidean relationships between cells, rather than operating on them as disconnected points in a

high-dimensional space. We therefore use this task to validate VBA-Transformer's core capabilities in implicit geometry discovery and effective representation learning.

**Data.** **Peripheral blood mononuclear cell (PBMC)**: We used the Zheng68K dataset, sequenced using the 10X Chromium platform, to train and evaluate the model for cell type annotation (Zheng et al., 2017). This large-scale scRNA-seq dataset contains 65,943 cells spanning 11 cell populations and includes expression profiles for 20,387 genes. The dataset was randomly split into 70% for training, 15% for validation, and 15% for testing. **Pancreatic datasets**: Pancreatic datasets were obtained from five studies: Baron (GSE84133)(Baron et al., 2016), Muraro (GSE85241)(Muraro et al., 2016), Xin (GSE81608)(Xin et al., 2016), Segerstolpe (E-MTAB-5061)(Segerstolpe et al., 2016), and Lawlor (GSE86473) (Lawlor et al., 2017). Among these, the Baron and Muraro datasets were used for training, while the remaining three datasets were used for testing. The training set contains 10,600 cells spanning 15 cell types, and the test set includes 4,218 cells representing 11 cell populations. **The human cell landscape (HCL) dataset** Han et al. (2020) comprises 599,926 cells across 59 tissues. This data is used for pretraining.

**Experimental Settings.** We benchmarked the VBA-Transformer model for single cell RNA sequencing (VBA-SC) against seven widely adopted single-cell models. Among them scGPT (Cui et al., 2024), scBERT (Yang et al., 2022), Geneformer(Theodoris et al., 2023) and TOSICA (Chen et al., 2023b) are transformer-based single-cell models. Seurat (Hao et al., 2021) and singleR (Aran et al., 2019) are correlation-based models. scNym (Kimmel & Kelley, 2021) is a recently proposed semi-supervised learning method. We evaluate annotation performance using two primary metrics: overall accuracy and the macro F1-score. Accuracy measures the proportion of correctly labeled cells, providing a general assessment of performance. Crucially, the macro F1-score is included to evaluate the model's ability to identify all cell types equally, giving more weight to the correct classification of rare cell populations, which is a key challenge in scRNA-seq analysis.

**Result.** The quantitative results for cell type annotation, presented in Table 2, provide a two-fold validation of our model's effectiveness. First, we evaluated the standard VBA-SC model trained from scratch. Even without the benefit of large-scale pre-training, VBA-SC demonstrates clear architectural superiority over other Transformer-based models. This is particularly evident on the challenging PBMC dataset, where it outperforms pre-trained methods such as scBERT and scGPT by a notable margin of over 3% in accuracy. This result highlights the power and data-efficiency of our model's geometric inductive bias. Next, to create a direct comparison with self-supervised methods, we introduced VBA-SC (SSL), which incorporates a pre-training stage. With the ability to first learn a general-purpose representation of the cellular manifold, our model's performance was substantially boosted, achieving new state-of-the-art results on both datasets. On the Pancreas dataset, VBA-SC (SSL) surpasses the strong Seurat baseline, reaching 97.64% accuracy. On the PBMC dataset, it further extends its lead over all competitors, achieving 80.68% accuracy.

This two-tiered success is particularly significant. The strong performance of the base model validates the power of our architecture, while the state-of-the-art results of the pre-trained version show that this superior architecture also effectively leverages self-supervised learning. This confirms that the core mechanism of VBA, identifying and leveraging the intrinsic geometric manifold of the data, is the main driver of its performance, enabling more accurate discrimination of fine-grained cell types and setting a new benchmark for this task.

## 4.4 EXPERIMENTS ON POINT CLOUD

The core thesis of our work is that VBA-Transformer is a generalizable model for geometric learning, whose principles are not confined to any single domain. To substantiate this claim, we must benchmark it on a canonical geometric task outside of bioinformatics to demonstrate its true universality. To this end, we apply our model to the task of 3D point cloud classification, a canonical benchmark in the field of geometric deep learning and a direct test of a model's capacity for geometric representation learning. Unlike the bioinformatics tasks, the goal here is to infer a global object category from the 3D coordinates of the points. This experiment evaluates whether VBA's core mechanism, vector bundle attention, can effectively learn a global representation to distinguish complex 3D shapes.

Table 2: Performance comparison of VBA-SC with baseline methods for cell type annotation

| | PBMC | | Pancreas | |
|---|---|---|---|---|
| **Model** | Accuracy | F1 | Accuracy | F1 |
| scBERT | 75.52% | 0.61 | 93.59% | 0.85 |
| TOSICA | 73.65% | 0.59 | 92.37% | 0.84 |
| scGPT | 75.47% | 0.61 | 93.10% | 0.85 |
| Geneformer | 74.67% | 0.60 | 92.72% | 0.84 |
| scNym | 69.50% | 0.53 | 89.72% | 0.80 |
| singleR | 68.40% | 0.50 | 91.82% | 0.84 |
| Seurat | 54.50% | 0.37 | 96.37% | 0.89 |
| **VBA-SC** | **78.71%** | **0.63** | **93.89%** | **0.86** |
| **VBA-SC (SSL)** | **80.68%** | **0.65** | **97.64%** | **0.89** |

**Data.** We evaluate our model on the canonical 3D point cloud classification task using the Model-Net40 benchmark (Wu et al., 2015). This dataset contains 12,311 clean 3D CAD models across 40 categories, such as airplane, sofa, plant, and desk. We follow the official split, with 9,843 models for training and 2,468 for testing. For each model, we uniformly sample 1,024 points from the object surface and use only their 3D coordinates as input, providing a direct test of geometric feature learning.

**Experimental Settings.** This experiment serves as a crucial test of VBA-Transformer's universality as a foundational geometric learning model. Our aim is not to outperform highly-specialized, fine-tuned architectures, but to demonstrate that our model's core principles can achieve strong, competitive results on a canonical task far outside its primary application domains. We compare VBA-Transformer against a suite of representative point cloud models, from the pioneering Point-Net(Qi et al., 2017a) and PointNet++(Qi et al., 2017b) to modern graph-based (DGCNN)(Wang et al., 2019) and attention-based methodsGuo et al. (2021); Zhao et al. (2021b); Han et al. (2022); Zhou et al. (2024); Chen et al. (2025). For a fair comparison, we adopt standard data augmentation and training procedures for ModelNet40. We report two key metrics: Overall Accuracy (OA) for a top-level performance summary, and mean-class Accuracy (mAcc) to ensure performance is robust and not biased towards majority object classes.

**Result.** As presented in Table 3, the results strongly support our central thesis on the universality of the VBA-Transformer. The goal was not to establish a new state-of-the-art, but to validate that our foundational architecture could generalize to a completely different domain. Our VBA-Transformer for point cloud (VBA-P) model's robust 92.9% OA confirms its effectiveness. More strikingly, the compact VBA-P-Tiny remains competitive with classic methods while operating at a scale comparable to highly efficient models, such as PointNet++. The significance of these findings lies not in breaking records but in the demonstration of true architectural versatility. The ability of VBA-Transformer, without domain-specific fine-tuning, to achieve such competent performance on this canonical benchmark validates its principles as a genuinely general-purpose framework for geometric learning.

## 4.5 ABLATION STUDY

To rigorously validate the architectural design of VBA-Transformer and quantify the contribution of its core components, we conducted a comprehensive ablation study across our Point Cloud, PBMC, and Pancreas datasets. We systematically replaced our proposed Vector Bundle Attention with several alternative mechanisms, with the results summarized in Table 4.

First, we confirmed the critical role of incorporating geometric information. The Transformer baseline, which is geometry-agnostic, shows a marked performance degradation across all datasets compared to its geometry-aware counterparts. For instance, its Overall Accuracy on the Point Cloud benchmark is nearly 3.2% lower than the Geometry-biased transformer (GBT) model(Venkat et al., 2023).

Table 3: Performance and efficiency comparison on the ModelNet40 benchmark.

| Model | Param.(M) | FLOPs(G) | OA(%) | mAcc(%) |
|---|---|---|---|---|
| PointNet | 3.5 | 0.4 | 89.2 | 86.2 |
| PointNet++ | 1.5 | 1.7 | 92.5 | 89.7 |
| DGCNN | 1.8 | 2.4 | 92.7 | 90.4 |
| PCT | 2.9 | 2.3 | 93.2 | 90.0 |
| GTNet | 2.1 | 4.3 | 93.2 | 92.6 |
| PointTransformer | 17.1 | 9.1 | 93.7 | 90.6 |
| DTNet | 6.4 | 4.4 | 93.4 | 90.9 |
| PointGA | 1.7 | 2.0 | 93.8 | 90.9 |
| VBA-P | 7.0 | 12.2 | 92.9 | 90.3 |
| VBA-P-Tiny | 1.7 | 2.9 | 90.1 | 87.2 |

Table 4: Ablation study of VBA-Transformer across three distinct datasets.

| Model | Point cloud | | PBMC | | Pancreas | |
|---|---|---|---|---|---|---|
| | OA(%) | mAcc(%) | Accuracy | F1 | Accuracy | F1 |
| **VBA-T** | **92.9** | **90.3** | **78.71%** | **0.63** | **93.89%** | **0.86** |
| Transformer | 82.6 | 78.2 | 66.76% | 0.44 | 88.92% | 0.71 |
| GBT | 85.8 | 81.7 | 70.32% | 0.56 | 90.33% | 0.83 |
| MQA-T | 83.1 | 78.9 | 67.73% | 0.48 | 87.33% | 0.67 |
| GQA-T | 83.6 | 79.1 | 66.57% | 0.43 | 87.19% | 0.66 |

The central finding is the superiority of our vector bundle paradigm. The GBT model, which incorporates geometry with a standard additive bias, is consistently and substantially outperformed by our full VBA-T model. On the Point Cloud benchmark, its overall accuracy is 7.1% higher, and on the PBMC dataset, the accuracy gap is over 8%. This demonstrates that simply adding a geometric bias is insufficient; VBA-T's ability to disentangle and operate on the geometric structure as a vector bundle provides a fundamentally more powerful representation.

Finally, we compared VBA-T to models using efficient attention variants, Multi-Query Attention (MQA-T)(Shazeer, 2019) and Grouped-Query Attention (GQA-T)(Ainslie et al., 2023). Despite these mechanisms offering improved computational efficiency, they perform significantly worse than VBA-T. This indicates that our model's performance gains stem from its unique geometric formulation rather than the specifics of the underlying standard attention implementation. Taken together, these ablations provide conclusive evidence that our proposed Vector Bundle Attention is a more effective mechanism for learning on geometric data than existing alternatives. More details shows in Appendix B.6

## 5 CONCLUSION

We introduce the VBA Transformer, a principled yet practical framework that treats attention as an operator on a learned geometric space. Its core, Vector Bundle Attention, disentangles base manifold geometry from fiber features. We enforce endpoint conditioned isometric transport to align fibers before similarity, embedding this geometric property directly in attention. A curvature informed correction from a learnable connection field captures nonflat manifold structure. Across domains, it is effective: state of the art on single cell RNA sequencing, competitive in spatial transcriptomics, and strong on 3D point clouds. Overall, the design pairs a hard isometry constraint with flexible, data driven parameterization, gaining geometric bias without full path integration. These results support geometric disentanglement and point to architectures with stronger, provable guarantees, for example equivariant and connection consistent transport, further narrowing the gap between deep learning and differential geometry.

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

## A APPENDIX I: DETAIL OF MODEL

### A.1 ADDITIONAL THEORETICAL CLARIFICATIONS

This section presents the formal geometric foundations of Vector Bundle Attention (VBA). We provide structured statements—Lemma, Theorem, and Claim—to clarify how each theoretical result directly supports a specific component of the mechanism. All proofs rely on the properties of skew-symmetric generators, the definition of curvature on vector bundles, and the structure of the proposed attention operator.

#### A.1.1 ISOMETRIC TRANSPORT

[Isometric Parallel Transport] Let $S_{\text{skew}}(b_i, b_j)$ be the skew-symmetric generator predicted from base coordinates. The transport operator

$$T_{j \to i} = \exp(S_{\text{skew}}(b_i, b_j)) \tag{9}$$

belongs to the special orthogonal group $SO(d_f)$ for all token pairs $(i, j)$. Thus, for any fiber vectors $u, v \in \mathbb{R}^{d_f}$,

$$\langle T_{j \to i} u, \ T_{j \to i} v \rangle = \langle u, v \rangle, \qquad \|T_{j \to i} u\| = \|u\|. \tag{10}$$

**Connection to the mechanism.** This result justifies the use of the exponential map: because $T_{j \to i}$ is a pure rotation, aligning keys and values prior to computing attention ensures that similarity is evaluated in a geometry-consistent fiber coordinate system.

#### A.1.2 CURVATURE-INDUCED FIBER MODULATION

[Curvature-Based Feature Adjustment] Let $\Gamma(b)$ denote the learned connection field over the base manifold, and let

$$\Omega(b) = d\Gamma(b) + \Gamma(b) \wedge \Gamma(b) \tag{11}$$

be the induced curvature 2-form. Define the effective curvature operator $R_{\text{eff}}(b)$ as a contraction of $\Omega(b)$. Then the curvature-aware fiber update

$$f_i \leftarrow f_i + R_{\text{eff}}(b_i) f_i \tag{12}$$

provides a first-order correction capturing local deviation from flat geometry.

**Connection to the mechanism.** This establishes why curvature appears as a multiplicative modulation of fiber features: it compensates for non-flat geometric structure and enables the model to adaptively adjust representations based on intrinsic manifold shape.

### A.1.3 GEOMETRY-CONSISTENT ATTENTION

[Invariance of Transport-Then-Attend] For queries $Q_i$, keys $K_j$, and the transported keys

$$\widetilde{K}_{j \to i} = T_{j \to i} K_j, \tag{13}$$

the VBA attention score

$$e_{ij} = \frac{\langle Q_i, \ \widetilde{K}_{j \to i} \rangle}{\sqrt{d_f}} \tag{14}$$

is invariant to arbitrary orthogonal frame rotations applied at token $j$. That is, for any $R \in SO(d_f)$,

$$K'_j = R K_j \quad \implies \quad e'_{ij} = e_{ij}. \tag{15}$$

**Connection to the mechanism.** This theorem shows that attention in VBA is intrinsically geometric: similarity is computed in the aligned fiber at $i$, guaranteeing that local coordinate choices at $j$ do not affect attention computation. This differentiates VBA from positional biases or post-hoc geometric corrections used in prior Transformers.

### A.1.4 STABILITY UNDER PERTURBATIONS

[Stability of Learned Transport] Let $\delta b$ denote a perturbation in base coordinates. Because the exponential map is 1-Lipschitz in the neighborhood of the identity for skew-symmetric generators, the induced transport perturbation satisfies

$$\|T_{j \to i}(b + \delta b) - T_{j \to i}(b)\| \le C \|\delta b\|, \tag{16}$$

for some constant $C$ depending on the generator network.

**Connection to the mechanism.** This claim explains the empirical robustness of VBA's transport operator: small geometric perturbations (e.g., noise or resolution changes) cannot cause large deviations in the transport alignment, stabilizing the entire attention computation.

### A.1.5 SUMMARY

Together, these structured results demonstrate that VBA embeds geometry directly into the attention operator through (1) isometric transport, (2) curvature-informed modulation, and (3) geometry-invariant similarity evaluation. This establishes a principled connection between differential geometric concepts and the computational mechanism used in the model.

### A.1.6 THEORY–MECHANISM CORRESPONDENCE

To make the geometric foundations of VBA clearer, Table 5 summarizes how each differential-geometric concept maps directly to a neural operator in our attention module.

### A.1.7 NOTATION SUMMARY

For clarity, Table 6 summarizes all core symbols used in the method section.

## A.2 FROM CONTINUOUS BUNDLE OBJECTS TO DISCRETE PARAMETERIZATION

**Vector bundle decomposition.** We regard our representation as a (learned) trivializing atlas: each token $i$ has a base coordinate $b_i \in \mathbb{R}^{d_b}$ and a fiber $F_{b_i} \cong \mathbb{R}^{d_f}$. The projection maps are linear:

$$b_i = W_b x_i, \qquad f_i^{(m)} = W_f^{(m)} x_i. \tag{17}$$

The bundle selector outputs $\alpha_i = \text{softmax}(s(x_i))$ and we form the aggregated fiber $f_i = \sum_m \alpha_i^{(m)} f_i^{(m)}$. LayerNorm/gating are applied:

$$\bar{f}_i = \text{LN}\big(\sigma(g(x_i)) \odot \phi(\sum_m \alpha_i^{(m)} A^{(m)} f_i^{(m)})\big). \tag{18}$$

Table 5: Correspondence between core geometric concepts and their neural realizations in VBA.

| Theoretical Concept | Neural Implementation (VBA Module) | Role in Architecture |
|---|---|---|
| **Bundle section** $(p, v)$: point $p \in M$ on the base manifold and vector $v \in F_p$ in the fiber | Geometric token $\text{Token}_i = (b_i, f_i)$, where $b_i$ is the base coordinate (projected from input) and $f_i$ is the fiber feature | **Disentanglement**: separates global geometry $(b_i)$ from local semantic representation $(f_i)$ so geometry can guide attention |
| **Parallel transport** $P_\gamma$: $F_q \to F_p$: moves a vector along a path while keeping it parallel | Transport operator $T_{j \to i} = \exp(S_{\text{skew}})$, an orthogonal matrix predicted from $(b_i, b_j)$. Keys/values transported as $K_{j \to i} = T_{j \to i} K_j$, $V_{j \to i} = T_{j \to i} V_j$ | **Alignment before attention**: aligns features into the fiber at $i$ so attention is computed in a consistent tangent space |
| **Metric compatibility**: inner products preserved by the connection | Isometric constraint $T_{j \to i} \in \text{SO}(d_f)$ via skew-symmetric $S_{\text{skew}}$ | **Stable scores**: rotation-only transport preserves norms, making $A_{ij} \propto Q_i^\top K_{j \to i}$ geometrically meaningful |
| **Curvature 2-form** $\Omega$: deviation from flatness / path dependence | Curvature correction: $f_i \leftarrow f_i + R_{\text{eff}}(b_i) f_i$, where $R_{\text{eff}}$ is computed from learned curvature $\Omega$ | **Inductive bias**: curvature modulates features to adapt to non-flat manifold structure |

Table 6: Summary of notation used in the proposed VBA model.

| Symbol | Meaning |
|---|---|
| $x_i$ | Input feature of token $i$ |
| $(b_i, f_i)$ | Base coordinate $b_i$ and fiber feature $f_i$ (geometric token) |
| $d_b, d_f$ | Dimensions of base and fiber spaces |
| $Q_i, K_i, V_i$ | Query, key, and value vectors derived from $f_i$ |
| $T_{j \to i}$ | Transport operator from fiber at $j$ to fiber at $i$ |
| $T_{j \to i} = \exp(S_{\text{skew}}(b_i, b_j))$ | Learnable isometric transport parameterization |
| $T_{j \to i} \in \text{SO}(d_f)$ | Rotation constraint ensuring metric compatibility |
| $K_{j \to i}, V_{j \to i}$ | Transported key and value at location $i$ |
| $A_{ij} \propto Q_i^\top K_{j \to i}$ | Attention score after geometric alignment |
| $\Omega$ | Curvature 2-form derived from learned connection |
| $R_{\text{eff}}(b_i)$ | Effective curvature operator at base point $b_i$ |
| $f_i \leftarrow f_i + R_{\text{eff}}(b_i) f_i$ | Curvature-modulated fiber update |

**A Geometrically Constrained Transport Operator.** In differential geometry, a connection defines the rules for parallel transport, which describes how a vector is transported along a path on a base manifold. A key property in many geometric spaces is isometry, meaning that the vector's length is preserved. To create a learnable operator that is both powerful and geometrically faithful, we design our `TransportNet` to structurally enforce this constraint. Specifically, we require the learned operator $T_{j \to i}$ to be a special orthogonal matrix ($T^\top T = I$, $\det(T) = 1$), ensuring that it performs a pure rotation in the fiber space.

This is achieved by parameterizing the operator via the matrix exponential. Given base coordinates $(b_i, b_j)$, `TransportNet`, a lightweight MLP, outputs a generator matrix $S \in \mathbb{R}^{d_f \times d_f}$, which is projected to the skew-symmetric space:

$$S_{\text{skew}} = \tfrac{1}{2}(S - S^\top). \tag{19}$$

The final operator is then defined as

$$T_{j \to i} = \exp(S_{\text{skew}}), \tag{20}$$

which guarantees that $T_{j \to i} \in SO(d_f)$ by construction.

We initialize the final layer of `TransportNet` with near-zero weights, so that $S_{\text{skew}} \approx 0$ at the start of training. This makes $T_{j \to i} \approx I$, stabilizing early optimization.

**Curvature Proxy.** Our model moves beyond simple algebraic proxies to compute a high-fidelity, position-dependent approximation of the formal curvature 2-form, $\Omega = d\Gamma + \Gamma \wedge \Gamma$. This section

provides a detailed breakdown of this calculation. The entire process is enabled by our learnable connection field, implemented as a neural network, `ConnectionNet`, which maps any base coordinate $b$ to the corresponding connection coefficient matrices for each basis direction, $\{\Gamma_k(b)\}$.

**The Algebraic Term ($\Gamma \wedge \Gamma$): Non-Commutativity of Transport.** The $\Gamma \wedge \Gamma$ term captures the quintessential feature of curvature: the path-dependence of parallel transport. Transporting a vector around an infinitesimal parallelogram by first moving in direction $i$ then $j$ is not the same as moving in direction $j$ then $i$. This failure to commute is the essence of curvature. In our discrete setting, this is captured by the commutator of the connection matrices for pairs of directions. For a 2D base manifold, the component is:

$$(\Gamma \wedge \Gamma)_{12} = [\Gamma_1(b), \Gamma_2(b)] = \Gamma_1(b)\Gamma_2(b) - \Gamma_2(b)\Gamma_1(b). \tag{21}$$

This term is computed directly from the outputs of the `ConnectionNet` at a given point $b$.

**The Derivative Term ($d\Gamma$): Inhomogeneity of the Geometric Structure.** The $d\Gamma$ term, or the exterior derivative of the connection, captures how the rules of transport themselves change from point to point. A non-zero $d\Gamma$ implies that the geometry is not homogeneous; the way vectors are transported at point $b$ is different from the way they are at a nearby point $b + \epsilon$. This is calculated from the partial derivatives of the connection coefficient matrices. For a 2D base manifold, the component is:

$$(d\Gamma)_{12} = \frac{\partial \Gamma_2}{\partial x_1} - \frac{\partial \Gamma_1}{\partial x_2}. \tag{22}$$

In our implementation, these partial derivatives are the elements of the Jacobian of the `ConnectionNet`'s output with respect to its input coordinates $b$. We compute this Jacobian efficiently using automatic differentiation.

**The Full Curvature Tensor and Practical Considerations.** The two components are combined to form the full, position-dependent curvature tensor, $\Omega(b)$, whose components are $\Omega_{ij}(b) = (d\Gamma)_{ij} + (\Gamma \wedge \Gamma)_{ij}$. This tensor provides a rich, local description of the learned manifold's geometry that is used to modulate the fiber representations.

This direct computation represents a deliberate trade-off between theoretical rigor and computational cost. Calculating the Jacobian of the `ConnectionNet` for a batch of points is computationally intensive. However, we argue this is a worthwhile investment for tasks where capturing the precise local geometry is critical, and it demonstrates the feasibility of directly integrating core objects of differential geometry into a network's architecture.

### A.3 ENDPOINT-DEPENDENT TRANSPORT VS. FORMAL PATH-DEPENDENT PARALLEL TRANSPORT

A frequent source of confusion is the term *path-dependent*. In differential geometry, parallel transport along a connection is intrinsically path-dependent: for points $b_j \to b_k \to b_i$ on a curve $\gamma$, the operator satisfies the **composition law** $T_{k \to i} \circ T_{j \to k} = T_{j \to i}$, where each $T$ is the path-ordered exponential of the connection integrated *along the specific path segment*. Nonzero curvature further implies nontrivial holonomy around loops.

**Our endpoint-dependent surrogate.** For tractability, our `TransportNet` maps endpoints to a transport operator,

$$T_{j \to i} = \text{TransportNet}(b_i, b_j) \in SO(d_f), \tag{23}$$

without explicitly integrating a connection along a path. This design does not enforce the composition law and therefore is *not* a faithful implementation of path-dependent parallel transport. Instead, it should be understood as an *endpoint-dependent surrogate* that summarizes the effect of transporting along an implicit canonical path between $b_j$ and $b_i$. We do not assume or compute geodesics; any reference to a 'canonical path' is descriptive rather than algorithmic.

**Rationale and implications.** Relaxing the composition law yields a lightweight, expressive operator that remains geometry-aware ($SO(d_f)$ parameterization and base coordinates) while avoiding the cost of path integration . The trade-off is that properties tied to true path dependence, such as

exact compositionality and holonomy derived from a single underlying connection, are not guaranteed. We find that this surrogate suffices for our tasks, but we view efficient approximations to genuine path integrals as promising future work.

## A.4 MATHEMATICAL DETAILS OF THE CURVATURE CALCULATION

Our model moves beyond simple algebraic proxies to compute a high-fidelity, position-dependent approximation of the formal curvature 2-form, $\Omega = d\Gamma + \Gamma \wedge \Gamma$. This section provides the detailed mathematical and implementation framework.

**The Learnable Connection Field.** The entire calculation is enabled by a learnable connection field, implemented as a neural network we term `ConnectionNet`. For a base manifold of dimension $d_b$, this network maps any base coordinate $b \in \mathbb{R}^{d_b}$ to a set of $d_b$ connection coefficient matrices, $\{\Gamma_k(b)\}_{k=1}^{d_b}$. Each matrix $\Gamma_k(b)$ represents the connection along the $k$-th coordinate direction and is an element of the general linear Lie algebra $\mathfrak{gl}(d_f, \mathbb{R})$, meaning it is a real-valued $d_f \times d_f$ matrix that acts on the fiber space.

**Generalization to Higher Dimensions** ($d_b > 2$). The full curvature 2-form $\Omega(b)$ is a tensor whose components, for any pair of directions $i$ and $j$, are given by the well-known formula:

$$\Omega_{ij}(b) = \frac{\partial \Gamma_j}{\partial x_i} - \frac{\partial \Gamma_i}{\partial x_j} + [\Gamma_i(b), \Gamma_j(b)]. \tag{24}$$

In a general $d_b$-dimensional space, there are $\binom{d_b}{2}$ such unique components. Our framework is fully general and can, in principle, compute this entire tensor. The derivative term involving the partial derivatives is computed from the Jacobian of the `ConnectionNet` via automatic differentiation. The algebraic term is computed from the commutator (Lie bracket) of the corresponding output connection matrices.

However, computing the full Jacobian and all $\binom{d_b}{2}$ commutator components for a high-dimensional base manifold is computationally prohibitive. Therefore, we clarify our practical implementation. For a general $d_b$, we compute a scalar summary of the total curvature by summing the magnitudes of a subset of the most significant components, or by learning a low-dimensional projection of the full tensor. For the 2D case presented for illustration, we compute the single component $\Omega_{12}(b)$ directly. This represents a pragmatic trade-off that retains the core principles of the formal theory while ensuring computational tractability.

## A.5 THEORETICAL GUARANTEES

In this section, we provide theoretical insights into the properties of our geometric attention mechanism. We begin by establishing a fundamental principle that an ideal geometric attention operator should satisfy, and then we analyze the stability of our practical implementation.

[**1. Invariance to Local Orthogonal Frame Changes**] Let a change of local basis in the fiber at each base point $b_p$ be represented by an orthogonal matrix $S_p \in O(d_f)$. An ideal transport operator should transform according to the rules of a geometric connection, i.e., $T'_{j \to i} = S_{b_i} T_{j \to i} S_{b_j}^{-1}$. If the fiber vectors (Queries, Keys) also transform accordingly ($Q'_i = S_{b_i} Q_i$, $K'_j = S_{b_j} K_j$), then the pre-softmax attention scores $e_{ij} = \langle Q_i, T_{j \to i} K_j \rangle$ are invariant.

*Proof.* By direct substitution,

$$
\begin{aligned}
\langle Q'_i, T'_{j \to i} K'_j \rangle &= (S_{b_i} Q_i)^\top \left( S_{b_i} T_{j \to i} S_{b_j}^{-1} \cdot S_{b_j} K_j \right) \\
&= Q_i^\top S_{b_i}^\top S_{b_i} T_{j \to i} K_j \\
&= Q_i^\top T_{j \to i} K_j = e_{ij},
\end{aligned}
\tag{25}
$$

where we used the orthogonality condition $S_{b_i}^\top S_{b_i} = I$. $\qquad \square$

**Remark on Theorem 1's Applicability.** Theorem 1 establishes a fundamental desirable property for any robust geometric attention mechanism: its output should not depend on the arbitrary choice of coordinate systems (bases) in each local fiber space. While our `TransportNet`, even when constrained to produce orthogonal transformations (isometries), is not guaranteed to satisfy the required equivariance property by construction, this theorem provides a strong theoretical motivation for its design. It also suggests a clear path for future work: designing `TransportNet` architectures that are explicitly equivariant to such transformations, which could lead to even better generalization.

[**2. Pre-softmax Score Perturbation Bound**]

Let the transport operator be parameterized residually as

$$T_{j \to i} = I + A_{j \to i}, \tag{26}$$

where $A_{j \to i}$ is initialized near zero. For fixed $Q_i$ and $K_j$, the change in the attention logit is bounded by

$$|e_{ij} - e_{ij}^{(0)}| \le \|Q_i\|_2 \, \|A_{j \to i}\|_2 \, \|K_j\|_2, \tag{27}$$

where $e_{ij}^{(0)} = \langle Q_i, K_j \rangle$.

**Proof.** We have

$$e_{ij} - e_{ij}^{(0)} = \langle Q_i, A_{j \to i} K_j \rangle. \tag{28}$$

By the Cauchy–Schwarz inequality,

$$|\langle Q_i, A_{j \to i} K_j \rangle| \le \|Q_i\|_2 \, \|A_{j \to i} K_j\|_2. \tag{29}$$

By the definition of the operator norm,

$$\|A_{j \to i} K_j\|_2 \le \|A_{j \to i}\|_2 \, \|K_j\|_2, \tag{30}$$

which gives the stated bound. □

**Remark on Consistency with the Exponential Implementation.** In practice, our transport operator is parameterized as the *matrix exponential* of a skew-symmetric generator,

$$T_{j \to i} = \exp(S_{\text{skew}}). \tag{31}$$

Near initialization, where $S_{\text{skew}}$ is close to zero, we can use the first-order approximation

$$\exp(S_{\text{skew}}) \approx I + S_{\text{skew}}, \tag{32}$$

which exactly matches the residual form assumed in Theorem 2. Therefore, the perturbation bound is directly applicable to the early stages of training, explaining why the initialization near the identity stabilizes optimization. As training progresses, higher-order terms in the exponential expansion enrich the expressivity of the operator, while the isometric property is preserved by construction.

A.6    Contraction and Invariance of the Curvature Tensor

Let $M$ be a $d_b$-dimensional base manifold with a Riemannian metric $g$ and $F \to M$ a rank-$d_f$ vector bundle modeling the fiber features. Given a learnable connection field (from `ConnectionNet`), we denote its curvature by $\Omega \in \Omega^2(M, \text{End}(F))$ with local components $\{\Omega_{ij}(b)\}_{1 \le i < j \le d_b}$, so $\Omega_{ij}(b) \in \mathbb{R}^{d_f \times d_f}$ and the full tensor lives in $\mathbb{R}^{d_b \times d_b \times d_f \times d_f}$. Our goal is to contract the base-space indices of $\Omega$ into a single corrective endomorphism $R_{\text{eff}}(b) \in \mathbb{R}^{d_f \times d_f}$ that modulates $\bar{f}_i$ via $\bar{f}_i \leftarrow \bar{f}_i + R_{\text{eff}}(b_i) \, \bar{f}_i$. We require that the construction be *invariant to orthogonal coordinate changes on the base* ($O(d_b)$), and stable numerically.

**A canonical $O(d_b)$-invariant contraction.** Changing orthonormal coordinates on $T_b M$ induces an $O(d_b)$ action on $\Lambda^2 T_b M$, thus the index pair $(i, j)$ mixes via an orthogonal matrix. A canonical, rotation-invariant contraction that removes the base indices and yields a fiber endomorphism is the *energy operator*

$$S(b) := \sum_{1 \le i < j \le d_b} \Omega_{ij}(b)^\top \, \Omega_{ij}(b) \in \mathbb{S}_+^{d_f}. \tag{33}$$

Under any $U \in O(d_b)$, the components $\{\Omega_{ij}\}$ mix orthogonally, hence $S(b)$ is unchanged: $S'(b) = \sum_{i<j} \Omega'_{ij}{}^\top \Omega'_{ij} = \sum_{i<j} \Omega_{ij}^\top \Omega_{ij} = S(b)$. If the connection is metric-compatible so that $\Omega_{ij} \in \mathfrak{so}(d_f)$, then $S(b) = -\sum_{i<j} \Omega_{ij}(b)^2$ is symmetric positive semidefinite (PSD).

**Invariant scalar features for learned gating.** While $S$ retains anisotropy in the fiber, it is also useful to summarize curvature strength by scalar invariants, which are invariant to both $O(d_b)$ and fiber-basis changes:

$$\kappa_1(b) = \operatorname{tr} S(b), \qquad \kappa_2(b) = \operatorname{tr}\big(S(b)^2\big), \qquad \kappa_3(b) = \operatorname{tr}\big(S(b)^3\big), \ldots \tag{34}$$

We additionally include the Frobenius norm of the full curvature tensor as a compact descriptor:

$$s(b) = \|\Omega(b)\|_F^2 = \sum_{i<j} \|\Omega_{ij}(b)\|_F^2 . \tag{35}$$

These scalars are fed into a lightweight MLP, dubbed `CurvatureAdapter`, to produce *scalar* gates used below.

**Directional contraction by a data-driven 2-form.** If a preferred local plane is available (such as, from a structure tensor on $M$), let $\Sigma(b) \in \Lambda^2 T_b M$ be a unit 2-form constructed in an $O(d_b)$-equivariant way (such as, take the top-2 eigenvectors of a base-space structure tensor, wedge them and $L^2$-normalize). Define the directional contraction

$$R_{\mathrm{dir}}(b) = \langle \Omega(b), \Sigma(b) \rangle = \sum_{i<j} \Sigma^{ij}(b)\, \Omega_{ij}(b) \in \operatorname{End}(F_b). \tag{36}$$

When the base coordinates rotate, both $\Omega$ and $\Sigma$ co-transform, and the inner product on $\Lambda^2$ is $O(d_b)$-invariant, hence $R_{\mathrm{dir}}$ is invariant to base rotations while retaining directional information.

**Unified effective curvature operator.** We combine the canonical invariant operator $S$ and the optional directional term $R_{\mathrm{dir}}$ using *invariant scalar gates* produced by `CurvatureAdapter`. Let $\kappa(b) = \big[s(b), \kappa_1(b), \kappa_2(b), \ldots\big]$ be the scalar feature vector. We define

$$\tilde{S}(b) = \frac{S(b)}{\operatorname{tr} S(b) + \varepsilon}, \qquad (\alpha, \beta, \gamma, \delta) = \texttt{CurvatureAdapter}\big(\kappa(b)\big), \tag{37}$$

$$R_{\mathrm{eff}}(b) = \alpha\, I + \beta\, \tilde{S}(b) + \gamma\, \tilde{S}(b)^2 + \delta\, R_{\mathrm{dir}}(b) . \tag{38}$$

Here $\varepsilon > 0$ stabilizes normalization; $I$ is the identity on the fiber. The use of only *scalar* gates preserves invariance: $\alpha, \beta, \gamma, \delta$ are functions of invariants and thus do not depend on the choice of coordinates.

**Special case $d_b = 2$.** When $d_b = 2$, there is a single curvature component $\Omega_{12}(b)$ up to sign, and $S(b) = -\Omega_{12}(b)^2$, $s(b) = \|\Omega_{12}(b)\|_F^2$. Then $R_{\mathrm{dir}}(b) = \pm\Omega_{12}(b)$ if $\Sigma$ is chosen as the (oriented) unit area 2-form. Hence equation 38 reduces to a polynomial in $\Omega_{12}^2$ with an optional linear term in $\Omega_{12}$.

**Invariance guarantees.** [Base-rotation invariance] Under any orthonormal change of base coordinates $U \in O(d_b)$: (i) $S$ in equation 33 is invariant; (ii) the scalars in equation 34 and equation 35 are invariant; (iii) if $\Sigma$ is constructed $O(d_b)$-equivariantly, $R_{\mathrm{dir}}$ in equation 36 is invariant. Consequently, $R_{\mathrm{eff}}$ in equation 38 is invariant to base rotations.

(i) follows from orthogonal mixing on $\Lambda^2$ and the sum-of-squares form; (ii) follows from the cyclicity of trace and invariance of $S$; (iii) follows since $\langle \cdot, \cdot \rangle$ on $\Lambda^2$ is $O(d_b)$-invariant and $\Omega, \Sigma$ co-transform. The scalar gates depend only on invariants, so equation 38 is invariant.

If $\Omega_{ij} \in \mathfrak{so}(d_f)$ (metric-compatible setting), then $S$ is PSD and diagonalizable with an orthonormal eigenbasis; any polynomial in $S$ is PSD and shares eigenvectors. Hence $I, \tilde{S}, \tilde{S}^2$ commute, yielding a numerically stable modulation. The scalars $\kappa_p = \operatorname{tr}(S^p)$ are invariant to fiber basis changes, and the gates $\alpha, \beta, \gamma$ preserve this invariance.

**Relation to the scalar-only adapter.** A purely scalar-driven approach would map $s(b)$ (or $\kappa$) directly to a matrix via an unconstrained MLP, $R_{\mathrm{eff}} = \texttt{MLP}(s)$.

Our unified design keeps the scalar *gating* but lets geometry enter through $S$ and $R_{\mathrm{dir}}$, giving both invariance and controllable anisotropy.

**Complexity and implementation notes.** Computing $S$ requires $d_b(d_b - 1)/2$ matrix products of size $d_f \times d_f$ per location (often shared across heads). In practice we: (i) accumulate $\sum_{i<j} \Omega_{ij}^\top \Omega_{ij}$ on-the-fly in FP16 with loss-scaling; (ii) cache $S$ per layer if curvature is reused by multiple heads; (iii) compute $\kappa_p$ via power iterations on $S$ to avoid explicit $S^p$ when $d_f$ is large.

**Backpropagation.** Let $L$ be the loss. Gradients through equation 33 follow from $\frac{\partial L}{\partial \Omega_{ij}} = \Omega_{ij}\left(\frac{\partial L}{\partial S} + \frac{\partial L}{\partial S}^\top\right)$, and through equation 38 by standard polynomial rules; for $R_{\text{dir}}$, $\frac{\partial L}{\partial \Omega_{ij}} \mathrel{+}= \delta \Sigma^{ij} \frac{\partial L}{\partial R_{\text{dir}}}$. The gates' gradients come by the `CurvatureAdapter` chain rule on $\kappa(b)$.

**Final update rule.** With $R_{\text{eff}}$ from equation 38, the feature update is

$$\bar{f}_i \leftarrow \bar{f}_i + R_{\text{eff}}(b_i)\, \bar{f}_i, \tag{39}$$

which is invariant to base rotations, stable under metric-compatible connections, and retains fiber-directional anisotropy by $S$ and $R_{\text{dir}}$ when used.

## A.7 TRIPLET CONSISTENCY OF ENDPOINT-CONDITIONED TRANSPORT

Our transport operator $T_{j\rightarrow i}$ is implemented as an endpoint-conditioned surrogate for true path-dependent parallel transport. To quantify how close it is to a composition-preserving connection, we evaluate a simple triplet consistency metric.

For randomly sampled triplets $(i, j, k)$, we define the Frobenius-norm discrepancy between composed and direct transports as

$$\Delta_{ijk} = \left\| T_{k\rightarrow i}\, T_{j\rightarrow k} - T_{j\rightarrow i} \right\|_F. \tag{40}$$

We report the average discrepancy

$$\mathbb{E}_{(i,j,k)}[\Delta_{ijk}] \tag{41}$$

over triplets drawn from a trained model.

On VBA-P (ModelNet40), the trained VBA model yields

$$\mathbb{E}_{(i,j,k)}[\Delta_{ijk}] = 0.05732, \tag{42}$$

indicating that the learned endpoint-conditioned transport behaves close to a composition-preserving parallel transport in practice.

## A.8 COMPLEXITY ANALYSIS

The computational complexity of our geometrically principled VBA layer is higher than that of standard attention, reflecting the trade-off for a more rigorous model. The primary costs arise from two new components: the constrained transport operator and the direct calculation of the curvature tensor.

**Transport Operator Complexity.** Our constrained `TransportNet` computes a unique orthogonal matrix for each of the $n^2$ token pairs. For each pair, this involves a forward pass through an MLP (cost $C_{\text{MLP}}$) to produce a generator matrix, followed by a matrix exponential (cost $C_{\text{exp}}$) to ensure orthogonality. The resulting $d_f \times d_f$ transport matrix is then applied to the key/value vectors (cost $O(d_f^2)$). The total complexity for the transport-attention stage is therefore:

$$O(n^2(C_{\text{MLP}} + C_{\text{exp}} + d_f^2)). \tag{43}$$

**Curvature Tensor Complexity.** The position-dependent curvature tensor $\Omega(b)$ is computed for each of the $N$ points. This requires evaluating the `ConnectionNet` (cost $C_{\text{ConnNet}}$), computing its Jacobian with respect to the base coordinates (a significant cost we denote as $C_{\text{Jacobian}}$), and calculating the commutator term (cost dominated by matrix multiplication, $O(d_f^3)$). The total per-layer cost for the curvature component is:

$$O(n \cdot (C_{\text{ConnNet}} + C_{\text{Jacobian}} + d_f^3)). \tag{44}$$

**Overall Complexity.** The overall complexity is dominated by the pairwise transport operator, which is significantly more intensive than the $O(n^2 d)$ complexity of standard self-attention. This highlights the deliberate design choice to prioritize theoretical guarantees over minimal computational cost.

**Efficiency–Accuracy Trade-offs** We additionally evaluate scalable variants of VBA using (1) local geometric attention and (2) a low-rank approximation of the transport operator. As shown in Table 7, both variants significantly reduce training time while maintaining competitive accuracy.

Table 7: Efficiency–accuracy trade-offs using local geometric attention and low-rank transport on the PBMC dataset.

| Method | Acc(%) | F1 | Time (s/epoch) |
|---|---|---|---|
| Full VBA | **78.71** | **0.63** | 876 |
| Sparse/Local Attention | 77.65 | 0.62 | 560 |
| Low-rank transport | 76.90 | 0.60 | 529 |

(1) Local geometric attention (top-$k$ neighbors). Because VBA learns explicit base coordinates, we can restrict transport and attention to each point's -nearest neighbors on the learned manifold. This reduces complexity from $O(n^2)$ to $O(nk)$.

On the PBMC scRNA-seq dataset, this approach achieves a 36% reduction in epoch time (from 876s to 560s) with only a 1.06-point accuracy drop (from 78.71 to 77.65). This strategy provides the best balance between efficiency and accuracy.

(2) Low-rank transport generators. Instead of predicting a full $d_f \times d_f$ generator matrix for the matrix exponential, we parameterize the generator in a low-rank form. This reduces the cost of both the exponential and the transport application. On PBMC, this yields a 40% speedup (from 876s to 529s) with a moderate accuracy trade-off (from 78.71 to 76.90).

## A.9 Algorithm Details

For clarity and reproducibility, we provide the detailed pseudocode for a single layer of our Vector Bundle Attention (VBA) mechanism in Algorithm 1. The algorithm outlines the three core stages discussed in the main paper: (1) the projection of input features into the vector bundle representation (base and fiber), (2) the computation of attention scores using the learnable, geometry-dependent parallel transport operator, and (3) the aggregation of transported value vectors to produce the final output.

## A.10 Differences from previous work

Our method represents a paradigm shift from *"Compare-then-Bias"* to *"Align-then-Compare"*.

Geometry inside attention. Prior models (such as Graphormer and GBT) compute dot-product similarity in the ambient feature space and add geometric information as an external bias term. In contrast, VBA aligns key–value features through a learned parallel-transport operator $T_{j \to i}$ before computing similarity. This makes geometry intrinsic to the attention operator rather than a post-hoc correction.

Intrinsic vector-bundle formulation. VBA defines attention on a learned base manifold and fiber space. The comparison $\langle Q_i, T_{j \to i} K_j \rangle$ takes place in a shared, geometry-consistent fiber. To our knowledge, no prior Transformer performs attention as an intrinsic geometric operator.

Pairwise orthogonal transport with curvature correction. VBA learns pairwise orthogonal transports and a curvature-based correction, enabling principled modeling of non-Euclidean biological and spatial manifolds. Earlier Euclidean or message-passing approaches do not provide this level of geometric expressiveness.

---

**Algorithm 1** The Vector Bundle Attention (VBA) Layer

---

1: **Parameters:** Projection matrices $W_b, W_f, W_q, W_k, W_v, W_o$; Learnable fields `ConnectionNet`($\cdot$), `TransportNet`($\cdot, \cdot$); Curvature scale $\lambda$.

2: **Input:** Sequence of features $X = \{x_i\}_{i=1}^N$, where $x_i \in \mathbb{R}^D$.

3: **Output:** Updated sequence $Y = \{y_i\}_{i=1}^N$.

4:             ▷ *1. Project inputs to initial base and fiber spaces.*

5: **for** $i = 1$ to $N$ **do**

6:   $b_i \leftarrow \text{LayerNorm}(W_b x_i)$         ▷ Base manifold coordinate

7:   $f_i \leftarrow \text{LayerNorm}(W_f x_i)$          ▷ Initial fiber feature

8:        ▷ *2. Apply Full Curvature Correction (O($d_b$)-invariant).*

9: **for** $i = 1$ to $N$ **do**

10:   $\Gamma(b_i) \leftarrow \text{ConnectionNet}(b_i)$       ▷ Connection 1-form at $b_i$

11:   $d\Gamma_i \leftarrow \text{Jacobian}_b(\text{ConnectionNet})(b_i)$    ▷ Autodiff w.r.t. base coords

12:   $\Omega_i \leftarrow d\Gamma_i + \Gamma(b_i) \wedge \Gamma(b_i)$     ▷ Curvature 2-form; $\wedge$ is commutator

13:   $S_i \leftarrow \sum_{p<q} \Omega_i[p,q]^\top \Omega_i[p,q]$   ▷ Energy operator (PSD), base-rotation invariant

14:   $\kappa_i \leftarrow \left[\text{tr}(S_i),\ \text{tr}(S_i^2)\right]$       ▷ Scalar invariants for gating

15:   $(\alpha_i, \beta_i, \gamma_i, \delta_i) \leftarrow \text{CurvatureAdapter}(\kappa_i)$    ▷ Scalar (coordinate-free) gates

16:   $\tilde{S}_i \leftarrow S_i / (\text{tr}(S_i) + \varepsilon)$      ▷ Stabilized normalization ($\varepsilon > 0$)

17:   **if** `useDirectional` **then**

18:    $\Sigma(b_i) \leftarrow \text{ComputeEquivariant2Form}(b_i)$   ▷ Equivariant unit 2-form from structure tensor

19:    $R_{\text{dir},i} \leftarrow \langle \Omega_i, \Sigma(b_i) \rangle$      ▷ Directional contraction in $\text{End}(F)$

20:   **else**

21:    $R_{\text{dir},i} \leftarrow 0$

22:   $R_{\text{eff},i} \leftarrow \alpha_i I + \beta_i \tilde{S}_i + \gamma_i \tilde{S}_i^2 + \delta_i R_{\text{dir},i}$    ▷ Final corrective operator

23:   $f_i \leftarrow f_i + \lambda R_{\text{eff},i} f_i$   ▷ Fiber modulation; invariant & numerically stable (PSD-based)

24:      ▷ *4. Compute attention using geometrically constrained parallel transport.*

25: **for** $i = 1$ to $N$ **do**          ▷ For each query token

26:   **for** $j = 1$ to $N$ **do**         ▷ For each key token

27:        ▷ *— Geometrically Constrained Transport —*

28:    $S_{j \to i} \leftarrow \text{MLP}_{\text{transport}}(b_i, b_j)$     ▷ Predict a generator matrix

29:    $S_{\text{skew}} \leftarrow \frac{1}{2}(S_{j \to i} - S_{j \to i}^\top)$     ▷ Enforce skew-symmetry

30:    $T_{j \to i} \leftarrow \exp(S_{\text{skew}})$   ▷ Compute orthogonal transport via matrix exponential

31:        ▷ *— Transported Attention Score —*

32:    $\tilde{K}_{j \to i} \leftarrow T_{j \to i} K_j$      ▷ Apply transport to the key vector

33:    $e_{ij} \leftarrow (Q_i^\top \tilde{K}_{j \to i}) / \sqrt{d_f}$

34:   $\{\alpha_{ij}\}_{j=1}^N \leftarrow \text{softmax}_j(\{e_{ij}\}_{j=1}^N)$

35:      ▷ *5. Aggregate transported values and produce final output.*

36:   $y_i^{\text{fiber}} \leftarrow \sum_{j=1}^N \alpha_{ij}(T_{j \to i} V_j)$    ▷ Apply transport to values and aggregate

37:   $y_i \leftarrow x_i + W_o y_i^{\text{fiber}}$      ▷ Project back and add residual

38: **return** $Y$

---

# B APPENDIX II: ADDITIONAL EXPERIMENT.

## B.1 BENCHMARKER SETTINGS

For all baseline methods, we adhered strictly to the best practices and official implementations from their original publications to ensure a fair and rigorous comparison. Preprocessing followed each method's official pipeline. Where available for any baseline, we used the exact hyperparameters published by the original authors. In cases where optimal settings were not provided for a specific dataset, we performed a search over our predefined parameter ranges. The configuration achieving the best performance on a strictly separate, held-out validation set was then selected for the final evaluation on the test set. At no point was the validation data used for training, or the test data used for model selection. For these pretrained Transformer baselines (scBERT and scGPT), we directly use the officially released pretrained weights provided by the original authors: scBERT: Pre-

Table 8: Comparison on the human breast cancer ST dataset.

| Metric | DiffusionST | GraphST | VBA-ST |
|--------|-------------|---------|--------|
| ARI    | 0.57        | 0.53    | 0.59   |
| AMI    | 0.56        | 0.55    | 0.68   |
| NMI    | 0.54        | 0.54    | 0.69   |
| COMP   | 0.61        | 0.64    | 0.69   |
| HOMO   | 0.62        | 0.63    | 0.69   |

trained weights from [`https://drive.weixin.qq.com/s?k=AJEAIQdfAAoUxhXE7r`]. scGPT: Pretrained weights from [`https://drive.google.com/drive/folders/1oWh_-ZRdhtoGQ2Fw24HP41FgLoomVo-y`]. These models are then fine-tuned on our dataset. For the fine-tuning stage, we employed a consistent training setup across all transformer-based models to ensure a fair comparison of their architectural adaptability.

When we pretrain scGPT and scBERT on exactly the same HCL subset as VBA-SC, their performance on PBMC does not improve and even slightly degrades (scGPT decreases from 75.47% to 74.42%, scBERT decreases from 75.52% to 74.16%), whereas VBA-SC with the same HCL data remains several points ahead. This suggests that the performance gap is mainly due to the geometric attention mechanism rather than to differences in pretraining data.

## B.2 ADDITIONAL RESULT OF VBA-ST

To provide a more comprehensive evaluation of clustering performance, we supplement the Adjusted Rand Index (ARI) results from the main paper with visualizations of Normalized Mutual Information (NMI) and Completeness scores across all 12 DLPFC samples. These metrics offer alternative perspectives on the quality of the identified spatial domains.

The results are presented in Figure 2. As shown, the performance trends observed in these figures are highly consistent with the ARI results. Our model, VBA-ST, demonstrates strong overall performance, frequently achieving highly competitive or state-of-the-art scores. This further confirms its ability to learn meaningful spatial representations and validates the conclusions drawn in the main paper.

To provide a qualitative assessment of our model's performance, we present a visual comparison of the clustering results on all 12 human dorsolateral prefrontal cortex (DLPFC) samples. Figure 3 displays the ground-truth anatomical annotations side-by-side with the spatial domains identified by our VBA-ST model in an unsupervised manner.

The visualizations show that VBA-ST is highly effective at recovering the known multi-layered cortical structures. The boundaries of the identified clusters align well with the ground-truth layers across most samples, confirming the quantitative results presented in the main paper. This visual evidence underscores our model's ability to learn meaningful, spatially coherent biological patterns from complex transcriptomics data.

To demonstrate that VBA-ST is not tailored to DLPFC only, we additionally evaluate on the 10x Genomics human breast cancer Visium dataset (`https://www.10xgenomics.com/datasets/human-breast-cancer-block-a-section-1-1-standard-1-1-0`). Using the same unsupervised clustering pipeline, VBA-ST outperforms DiffusionST and GraphST on all metrics (ARI, AMI, NMI, completeness, homogeneity), with especially large gains on information-theoretic and region-quality scores (Table 8). This confirms that our geometric inductive bias transfers to spatial tissues with very different morphology.

## B.3 ADDITIONAL RESULT OF VBA-SC

To provide a qualitative and visual assessment of our VBA-SC model's cell type annotation performance, we present visualizations on the PBMC and Pancreas datasets. To ensure a direct comparison, we use the t-SNE coordinates provided by the original source datasets for plotting. Figure 4

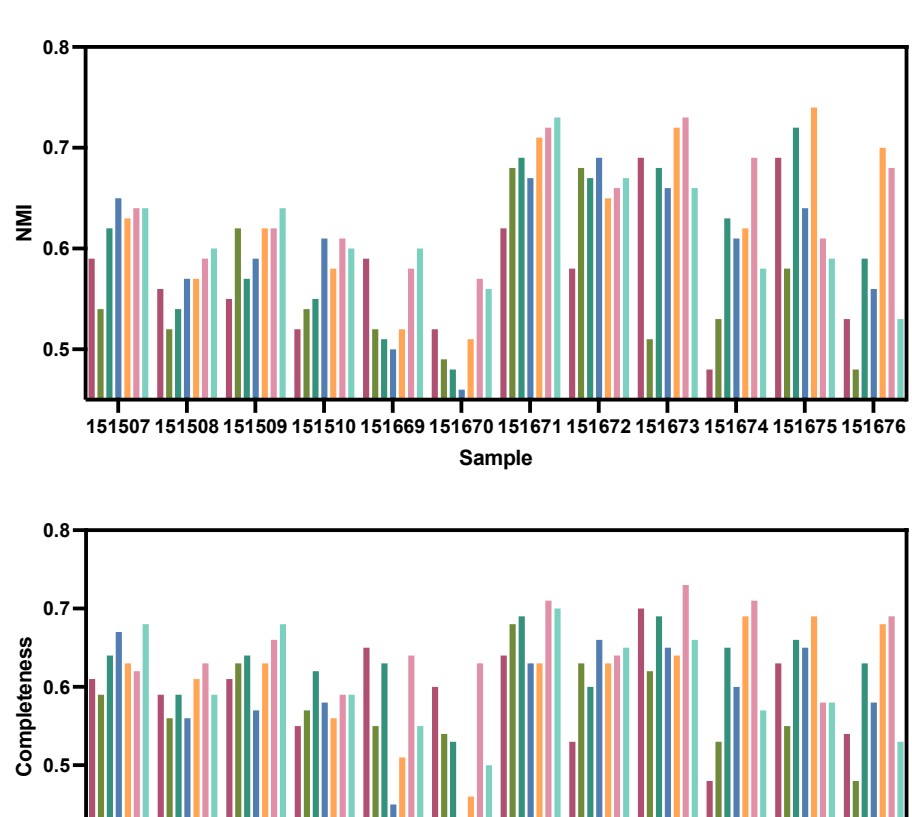

Figure 2: Comprehensive clustering performance on the 12 DLPFC samples, measured by NMI and Completeness. **(Top)** Visualization of Normalized Mutual Information (NMI) scores for all methods. **(Bottom)** Visualization of Completeness scores for all methods. These results corroborate the findings reported in the main text, highlighting the robust performance of VBA-ST.

shows a side-by-side comparison where cells are colored according to their ground-truth labels versus the labels predicted by our model.

A high degree of visual concordance can be observed between the predicted labels and the ground-truth annotations for both datasets. The spatial distribution of predicted cell types closely matches that of the ground truth, indicating a low misclassification rate. This provides strong visual support for the high Accuracy and F1-scores reported in the main text.

We report wall-clock time per epoch on PBMC for all major baselines using identical hardware and software configurations. As shown in Table 9, VBA is more computationally expensive but remains feasible for biological workloads and achieves substantially higher accuracy and F1 than all alternatives.

## B.4 ADDITIONAL RESULT OF VBA-P

**SO(3) Rotation Robustness** To evaluate the rotation robustness of VBA-P, we follow standard SO(3) evaluation protocols and assess the model under three settings:

z/z: trained and tested with upright orientations;

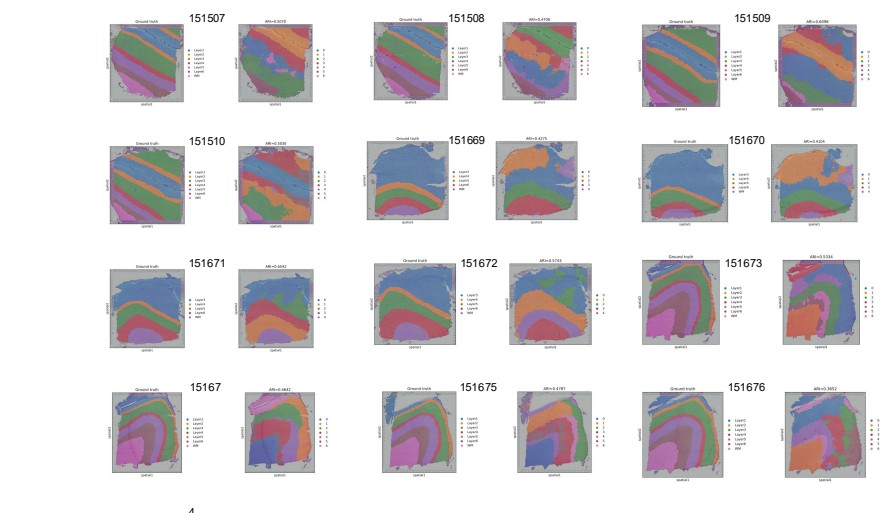

Figure 3: Clustering performance of our proposed VBA-ST

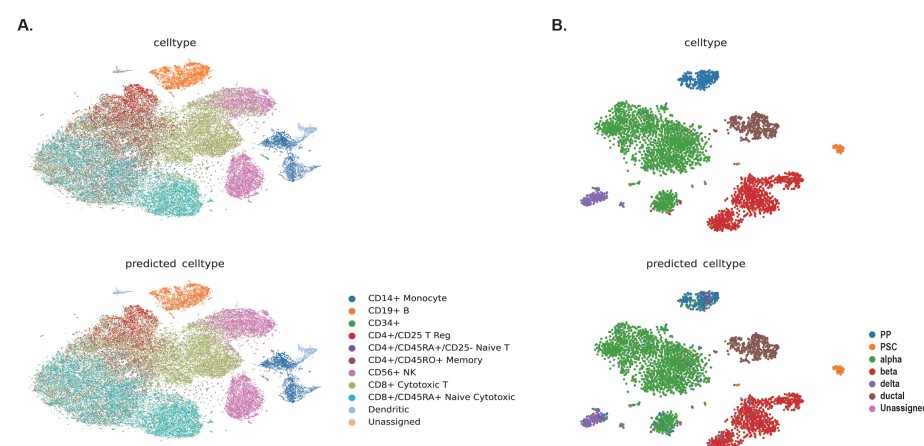

Figure 4: The t-SNE of gene expression of cells from the PBMC data (A) and the Pancreas Dataset (B). Up one is coloured by expert-annotated cell types from the original research. The down panel is colored by the prediction results of VBA-SC Model.

Table 9: Wall-clock time per epoch on the PBMC dataset, measured on identical hardware (4×GH200). VBA provides substantially higher accuracy despite a higher computational cost.

| Method | Accuracy (%) | F1 | Avg epoch time (s) |
|---|---|---|---|
| Transformer | 66.76 | 0.56 | 273 |
| GBT | 70.32 | 0.56 | 493 |
| MQA-T | 67.73 | 0.48 | 267 |
| GQA-T | 66.57 | 0.43 | 259 |
| **VBA (Ours)** | **78.71** | **0.63** | 876 |

z/SO(3): trained with upright orientations, tested with uniformly sampled SO(3) rotations;

SO(3)/SO(3): trained and tested with randomly sampled SO(3) rotations.

For each test shape, we apply ten independently and uniformly sampled SO(3) rotations and report the average accuracy.

Table 10: SO(3) rotation robustness on ModelNet40. VBA-P shows strong stability compared to attention-based and graph-based baselines.

| Method | z/z | z/SO(3) | SO(3)/SO(3) |
|---|---|---|---|
| VBA-T (Ours) | **92.9** | **77.9** | **83.9** |
| Transformer | 82.6 | 17.5 | 71.3 |
| GBT | 85.8 | 56.9 | 73.2 |
| MQA-T | 83.1 | 28.4 | 71.7 |
| GQA-T | 83.6 | 21.3 | 70.9 |

These results (in Table 10) show that VBA-P significantly reduces performance degradation under SO(3) perturbations, confirming that the "transport-then-attend" operator provides meaningful geometric robustness.

**ScanObjectNN: Real-world 3D Recognition** We additionally evaluate VBA-P on ScanObjectNN (PB-T50-RS split) (Uy et al., 2019), a challenging real-world benchmark with background clutter, occlusion, and viewpoint changes.

Table 11: Performance on ScanObjectNN (PB_T50_RS split). VBA-P is competitive with recent strong 3D point-cloud models.

| Model | Accuracy (%) |
|---|---|
| PointNet | 68.2 |
| PointNet++ | 77.9 |
| DGCNN (Phan et al., 2018) | 78.1 |
| PointGPT-S (Chen et al., 2023a) | 89.2 |
| PointMamba (Liang et al., 2024) | 89.3 |
| ACT (Dong et al., 2022) | 88.2 |
| **VBA-P (Ours)** | **81.7** |

Although VBA-P is not a point-cloud–specific model, it achieves strong performance. (Table 11).

### B.5 QM9 MOLECULAR PROPERTY PREDICTION

To assess whether VBA generalizes to molecular geometric learning, we conduct a preliminary experiment on the QM9 dataset. Atomic coordinates are treated as base manifold inputs and atom-level embeddings as fiber features. We train a lightweight VBA model to predict the $U_0$ property using the standard QM9 split.

VBA achieves a validation MAE of **11.49 meV**, which is competitive with widely used geometric baselines such as SchNet (14 meV) Schütt et al. (2017) and EGCN (11 meV) Lu et al. (2019). These results indicate that vector-bundle attention transfers effectively to molecular tasks without any architectural modifications.

Due to the limited time available during the rebuttal period, this experiment is not fully tuned; further optimization is expected to improve performance. We include this result here for completeness.

### B.6 MORE DETAILS IN ABLATION STUDY.

**Ablation of Architectural Components** To further investigate the contribution of the specific design choices that enhance the expressivity and geometric awareness of our model, we conducted a series of ablations on its advanced components. The experiments were performed on the Model-Net40 dataset, and the results are summarized in Table 12. The results clearly demonstrate that each of these components makes a valuable and distinct contribution to the model's overall performance. The most significant performance degradation is observed when reducing the Fiber Attention to the Scaled Dot-Product Attention of MHA. Furthermore, disabling the Curvature Correction leads to

a noticeable drop in accuracy, providing empirical evidence that this theoretically motivated component successfully helps the model capture the non-Euclidean nature of the data. To determine the relative importance of the curvature components, we conduct an ablation study. We evaluate a model equipped solely with the commutator term $(\Gamma \wedge \Gamma)$ and compare its performance to a model equipped only with the exterior derivative $(d\Gamma)$. Including the commutator term, $[\Gamma_i, \Gamma_j]$, results in a significant performance increase, achieving an Overall Accuracy 0.8% higher and a mean-class Accuracy 1.4% higher than the model with only the $d\Gamma$ term. This indicates that the model derives more benefit from capturing the path-dependent, non-abelian nature of the learned connection.

Table 12: Ablation study of key components in the VBA-Transformer on the ModelNet40 dataset.

| Model Variant | Projection | Curvature | Fiber | Connection | OA(%) | mAcc(%) | Avg epoch time (s) |
|---|---|---|---|---|---|---|---|
| VBA-T (Full Model) | ✓ | ✓ | ✓ | ✓ | **92.9** | **90.3** | **96.0** |
| w/o Projection | ✗ | ✓ | ✓ | ✓ | 86.1 | 81.9 | 83.4 |
| w/o Curvature | ✓ | ✗ | ✓ | ✓ | 87.6 | 82.3 | 61.0 |
| w/o $\Gamma \wedge \Gamma$ | ✓ | only $d\Gamma$ | ✓ | ✓ | 88.1 | 83.9 | 75.4 |
| w/o $d\Gamma$ | ✓ | only $\Gamma \wedge \Gamma$ | ✓ | ✓ | 88.9 | 85.3 | 70.5 |
| w/o Fiber | ✓ | ✓ | ✗ | ✓ | 85.7 | 82.4 | 87.8 |
| w/o Connection | ✓ | ✓ | ✓ | ✗ | 86.6 | 83.1 | 86.2 |

To assess which geometric components matter in noisy biological settings, we perform ablations on PBMC (Table 13). Removing the learned connection or curvature degrades accuracy by 5–7 percentage points, confirming that transport and curvature correction both contribute beyond simple geometry-aware encodings.

Table 13: Ablation study of key components in VBA on the PBMC dataset.

| Model Variant | Projection | Curvature | Fiber | Connection | Acc(%) | F1 | Avg epoch time (s) |
|---|---|---|---|---|---|---|---|
| Full VBA | ✓ | ✓ | ✓ | ✓ | **78.71** | **0.63** | **876** |
| w/o Projection | ✗ | ✓ | ✓ | ✓ | 72.94 | 0.58 | 690 |
| w/o Curvature | ✓ | ✗ | ✓ | ✓ | 73.28 | 0.60 | 474 |
| w/o $\Gamma \wedge \Gamma$ | ✓ | only $d\Gamma$ | ✓ | ✓ | 75.32 | 0.61 | 552 |
| w/o $d\Gamma$ | ✓ | only $\Gamma \wedge \Gamma$ | ✓ | ✓ | 75.53 | 0.62 | 498 |
| w/o Fiber | ✓ | ✓ | ✗ | ✓ | 71.87 | 0.59 | 774 |
| w/o Connection | ✓ | ✓ | ✓ | ✗ | 72.05 | 0.59 | 738 |

Beyond accuracy, we care about how expensive a model is to train. We adopt the Average time per epoch as the reference yardstick for training cost. This pattern indicates that the Jacobian-related $d\Gamma$ term is the main contributor to runtime overhead, while the commutator $\Gamma \wedge \Gamma$ is comparatively cheap.

**Hyperparameter Tuning and Analysis**  This section summarizes our hyperparameter tuning strategy and the final configurations used in all main experiments. We conducted extensive sensitivity analyses over the search space described in Table 14, where each key hyperparameter of the VBA-Transformer was systematically varied while the others were fixed to their optimal values from Table 15. Final hyperparameters for the three main tasks were chosen based on validation performance and are reported in Table 15. An exception was made for the **VBA-ST** model: to ensure fair comparison with baselines, its model dimension was fixed at 3,000, matching the input feature dimension used by competitor methods such as SpaGCN and DiffusionST.

## C  APPENDIX III: INTERPRETABILITY

A key advantage of the Vector Bundle Attention (VBA) architecture lies in its structured design, which offers a unique window into the model's internal workings. By decomposing each input

Table 14: Hyperparameter search space explored during model tuning and sensitivity analysis.

| Hyperparameter | Tested Values (Search Space) |
|---|---|
| Model Dimension ($D$) | $\{32, 64, 128, 256, 512\}$ |
| Number of Layers | $\{6, 8, 16\}$ |
| Number of Attention Heads | $\{4, 8, 16\}$ |
| Fiber Dimension ($d_f$) | $\{8, 16, 32, 64, 128\}$ |
| Number of Bundles ($M$) | $\{4, 8, 16, 32\}$ |
| Curvature Scale ($\lambda$) | $\{0.1, 0.2\}$ |
| Dropout Rate | $\{0.0, 0.1\}$ |

Table 15: Best hyperparameter configurations for the VBA-Transformer models used in the main experiments across the three tasks.

| Hyperparameter | VBA-ST | VBA-SC | VBA-P |
|---|---|---|---|
| Model Dimension ($D$) | 3000* | 512 | 128 |
| Number of Layers | 16 | 6 | 16 |
| Number of Attention Heads | 16 | 8 | 16 |
| Fiber Dimension ($d_f$) | 64 | 128 | 64 |
| Number of Bundles ($M$) | 32 | 4 | 32 |
| Curvature Scale ($\lambda$) | 0.1 | 0.1 | 0.1 |
| Dropout Rate | 0.0 | 0.1 | 0.1 |

*Set to match baselines for fair comparison, not tuned.

token into multiple fiber bundle representations, our model can learn specialized feature extractors. To investigate whether the model leverages this capability, we conducted an interpretability analysis by visualizing the learned bundle mixing weights.

## C.1 METHODOLOGY.

For each input point $x_i$ in a point cloud, the bundle selector network produces normalized weights $\alpha_i = \{\alpha_i^{(m)}\}_{m=1}^M$, where $\alpha_i^{(m)}$ quantifies the contribution of the $m$-th bundle to the final representation of that point. We visualize these weights in two ways:

1. **Spatial Mapping:** We color each point $i$ in the 3D point cloud according to its activation weight $\alpha_i^{(m)}$ for a specific bundle $m$. This allows us to observe which geometric regions of an object a particular bundle focuses on.

2. **Class-level Usage:** We compute the average bundle usage for an entire object class by averaging the weights $\{\alpha_i\}_i$ across all points of all instances within that class. This reveals which bundles are most important for identifying a particular object category.

## C.2 ANALYSIS OF LEARNED BUNDLE SPECIALIZATION.

Figure 5 presents our interpretability analysis for three instances across two object classes from the ModelNet40 dataset: one 'table' (A) and two distinct 'airplane' instances (B and C). The results reveal a clear and consistent pattern of learned bundle specialization, providing insight into the model's decision-making process.

First, we observe distinct bundle usage patterns between different classes. The 'desk' instance (A) relies on a different set of primary bundles compared to the 'airplane' instances (B and C), as shown in the bar charts. This indicates that the model successfully learns to activate different feature extractors for different object categories.

More strikingly, the analysis reveals a high degree of consistency within the same class. Both airplane instances (B and C), despite variations in their specific shapes, show the highest activation for the exact same bundle (Bundle 24). The spatial mapping confirms that this specific bundle

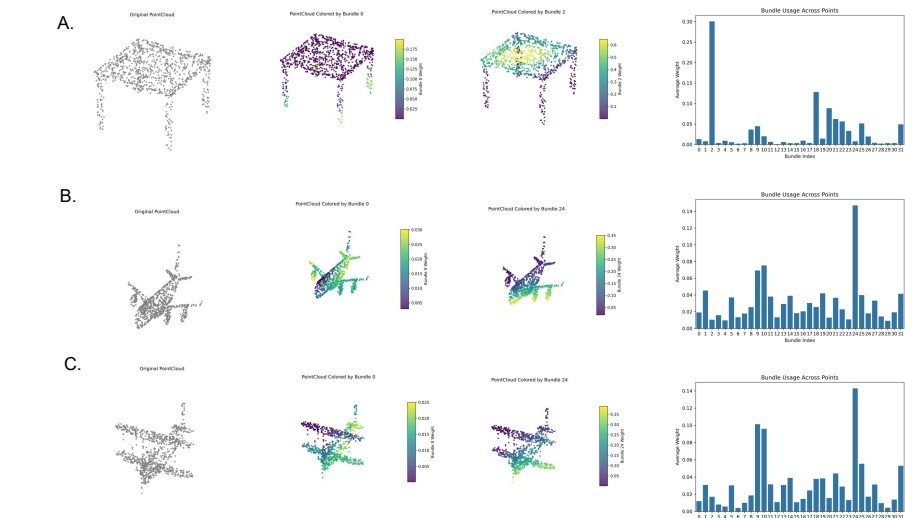

Figure 5: Visualization of learned bundle weights for a 'desk' (A) and two 'airplane' (B, C) instances, including Original point clouds, Spatial mapping of weights for the first bundle, Spatial mapping for the most-used bundle per instance, and Bar charts showing average bundle usage.

has learned to consistently focus on core structural elements of an airplane, such as the wings and fuselage. This intra-class consistency is powerful evidence that the bundles are not merely detecting random low-level features, but are learning to function as consistent, semantically meaningful feature extractors. This demonstrates that VBA-Transformer learns generalizable and interpretable representations, a key advantage of our structured geometric approach.

## C.3 Interpreting the Role of Curvature

To understand where the learnable curvature component plays a key role in our architecture, we visualized its effect on representative samples from the ModelNet40 test set. Specifically, for each point $\mathbf{p}_i$ on an object, we computed the L2 norm of the final curvature correction vector, $\|\delta_c(\mathbf{p}_i)\|_2$, which is applied to the point's feature representation within each VBA Block. This scalar value directly represents the intensity of the learned non-Euclidean adjustment just before the self-attention step.

Figure 6 shows this intensity mapped as a heatmap onto an 'airplane' sample, illustrating its evolution across six different VBA Blocks. We observe a clear and intuitive pattern: the magnitude of the learned correction is highest in regions of high geometric complexity, such as the wingtips, engines, and tail assembly. This effect becomes more pronounced in deeper layers, where the model has learned to segment the object into its primary structural parts.

This provides strong evidence that our model is not applying the curvature correction uniformly, but has instead learned an interpretable and meaningful representation of local geometric complexity. It leverages this understanding to apply stronger non-Euclidean adjustments only where geometrically justified, thereby adapting the feature space to the underlying structure of the data.

## C.4 Biological Relevance of Learned Features.

To assess whether our model learns biologically relevant representations, we investigated the feature weights driving the annotation of a specific cell type: CD56+ Natural Killer (NK) cells from the PBMC dataset. We identified the most salient genes contributing to the model's predictions for this population.

In particular, the identified gene set represents a canonical signature of highly activated, cytotoxic NK cells and shows a strong overlap with genes known to be upregulated during NK cell generation

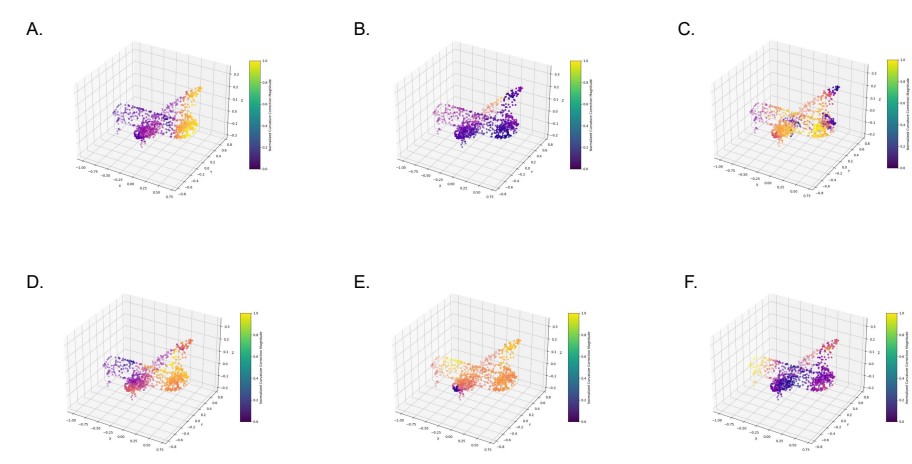

Figure 6: Visualization of the normalized curvature correction magnitude, $\|\delta_c(\mathbf{p}_i)\|_2$, for an 'airplane' sample across the six different `VBA Blocks` of the network. Yellow indicates high magnitude (strong feature correction), while purple indicates low magnitude. The progression from First Block (A) to Last Block (F) illustrates a clear hierarchical learning process.

and activation (Lehmann et al., 2012). The list is heavily enriched with genes encoding key components of the cytotoxic machinery, including GNLY, GZMA, GZMB, PRF1, and CCL4 (Figure 7).

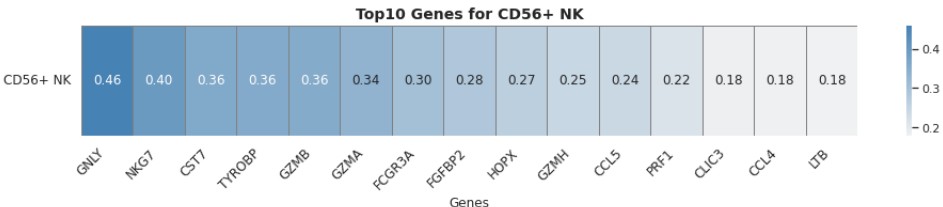

Figure 7: Feature importance scores of the top genes identified by VBA-SC as most predictive for the annotation of CD56+ Natural Killer (NK) cells.

This strong concordance with established immunological research demonstrates that VBA-SC is not operating as an uninterpretable black box. Instead, our model successfully learns to prioritize the biologically critical gene expression programs that define cellular identity and function. This ability to extract meaningful biological features underscores the effectiveness of our geometric representation learning approach.

## D  LIMITATIONS AND FUTURE WORK

While our work demonstrates the significant potential of the VBA-Transformer, we acknowledge several limitations that also highlight exciting avenues for future research.

First, the computational complexity of the VBA layer presents a scalability challenge. The pairwise application of our learnable transport operator results in a complexity of approximately $O(n^2 d_f^2)$, where $n$ is the sequence length and $d_f$ is the fiber dimension (Appendix A.8). This is notably higher than a standard Transformer's $O(n^2 d)$ complexity, as our method requires a matrix vector product for each of the $n^2$ pairs, rather than a more efficient dot product. This can make the model computationally demanding for extremely large-scale datasets. A promising direction for future work is to integrate principles from the efficient Transformer literature, such as sparse attention (by

defining local neighborhoods on the base manifold) or low-rank parameterizations of the transport operator, to mitigate this cost without sacrificing geometric fidelity.

Second, the choice of the base manifold dimension, $d_b$, is a key hyperparameter that governs the model's capacity to represent the underlying geometry. While our sensitivity analysis indicates that the model is robust within a reasonable range of values, identifying the optimal $d_b$ for a novel dataset currently relies on empirical tuning. Future research could explore methods for automatically adapting or even learning the intrinsic dimensionality directly from the data, which would enhance the model's autonomy and ease of use.

## E    LLM USAGE STATEMENT.

In this work, large language models (LLMs) were used solely as general-purpose assistive tools for grammar correction and language polishing. All technical content, research ideas, experiments, and analyses were conceived, designed, and executed solely by the authors. The LLM did not contribute to the scientific content, experimental design, or interpretation of the results.

## F    REPRODUCIBILITY STATEMENT

We have taken deliberate steps to ensure the reproducibility of our work. Detailed descriptions of the model architecture, training process, and hyperparameter settings are presented in the main text and supplementary materials. An anonymous implementation of the core model is also made available in the supplementary code.

