# OpenReview forum: "VBA: Vector Bundle Attention for Intrinsically Geometry-Aware Learning"
_ICLR.cc/2026/Conference — ICLR 2026 Conference Withdrawn Submission_

### Official Review · Reviewer_6PLt · 2025-10-30

**Soundness:** 2
**Presentation:** 2
**Contribution:** 3
**Rating:** 4
**Confidence:** 4

**Summary:**

This paper introduces a vector bundle attention transformer designed for learning representations from geometrically structured data across various fields, including biology, physics, and computer vision. A key innovation of this work is the redefinition of attention as an intrinsic geometric operator. Extensive experiments conducted on multiple benchmark datasets demonstrate the effectiveness of the proposed method.

**Strengths:**

1. This paper proposes a Vector Bundle Attention mechanism that operates attention on a learned geometric manifold.

2. The paper presents several theoretical analyses of the induced vector bundle attention.

3. A series of experiments are conducted on various datasets to demonstrate the superiority of the method across three different fields.

**Weaknesses:**

1. The main differences between the proposed attention mechanism and previous works are not clearly articulated in the current version.

2. The connections between the introduced theory and the proposed attention mechanism are somewhat unclear.

3. The notation and mathematical formulations could be significantly improved to help readers better understand the primary ideas presented in the paper.

**Questions:**

Apart from the identified weaknesses, I have the following questions that need to be addressed by the authors:

1. What is the main reason for introducing Curvature Correction in this paper? The authors are encouraged to provide a detailed explanation.

2. The total computational complexity of the method is related to the number of points $N$, with a complexity of $\mathcal{O}(N^2d)$. This raises concerns about the method's applicability to large-scale datasets. I suggest that the authors present some strategies to mitigate this limitation.

---

> ### Author Response · Authors · 2025-11-23
> **Authors’ Response to Reviewer 3 (Weakness 1&2)**
>
> We thank the Reviewer for the constructive feedback and for recognizing the novelty, theoretical grounding, and broad experimental scope of our work. Below, we address each of your comments in detail. We hope that these updates fully resolve your concerns and that you will consider revising your score.
>
> **W1 (Differences from previous work):**
> We thank you for this constructive feedback. We agree that the distinction between VBA and previous geometric attention mechanisms should be clearer. Our method represents a paradigm shift from "*Compare-then-Bias*" to "*Align-then-Compare*".
>
> * Geometry inside attention.
> Prior models compute dot-product similarity in the ambient feature space and add geometric information as an external bias term. In contrast, VBA aligns key–value features through a learned parallel-transport operator $T_{j \rightarrow i}$ before computing similarity. This makes geometry intrinsic to the attention operator rather than a post-hoc correction.
>
> * Intrinsic vector-bundle formulation.
> VBA defines attention on a learned base manifold and fiber space. The comparison
> ⟨$Q_i​$,$T_{j→i}​K_j$⟩ takes place in a shared, geometry-consistent fiber. To our knowledge, no prior Transformer performs attention as an intrinsic geometric operator.
>
> * Pairwise orthogonal transport with curvature correction.
> VBA learns pairwise orthogonal transports and a curvature-based correction, enabling principled modeling of non-Euclidean biological and spatial manifolds. Earlier Euclidean or message-passing approaches do not provide this level of geometric expressiveness.
>
> **W2 (Theory–mechanism connection):**
> We thank you for raising this point. We agree that making the link between our geometric theory and the implemented attention mechanism more explicit improves clarity. We added a triplet transport-consistency test (see W2 response to Reviewer gAi7). The small Frobenius-norm discrepancy shows that the learned endpoint-conditioned transport behaves consistently with the underlying geometric structure. This provides empirical evidence that the model internalizes the theoretical connection-based formulation.
>
> To further clarify how each theoretical component is realized in the architecture, we added the following mapping in the Appendix:
> Parallel transport → implemented through a learned skew-symmetric generator, whose matrix exponential yields an orthogonal operator in SO($d_f$).
>
> Fiber–base decomposition → implemented by projecting tokens into learned geometric coordinates (base) and fiber features.
>
> Curvature correction → implemented as a PSD operator constructed from $\Gamma$ and d$\Gamma$, providing rotation-invariant geometric modulation of the fiber.
>
> We also included a table summarizing this mapping. These revisions make the connection between theory and implementation explicit and show how each geometric concept directly shapes the design and behavior of vector-bundle attention.
> | Theoretical Concept | Neural Implementation (VBA Module) | Role in Architecture |
> |-|-|-|
> | **Bundle section** $(p, v)$ – a point $p \in M$ on the base manifold and a vector $v \in F_p$ in the fiber | **Geometric token** $\text{Token}_i = (b_i, f_i)$. Here $b_i$ is the base coordinate (projected from the input) and $f_i$ is the fiber feature (after bundle mixing). | **Disentanglement.** Separates global manifold structure $(b_i)$ from local semantic representation $(f_i)$, allowing geometry to explicitly guide attention. |
> | **Parallel transport** $P_\gamma : F_q \to F_p$ – moving a vector along a path while keeping it “parallel” | **Transport operator** $T_{j \to i} = \exp(S_{\text{skew}})$, an orthogonal matrix predicted from $(b_i, b_j)$ by TransportNet. Keys and values are transported as $K_{j \to i} = T_{j \to i} K_j$ and $V_{j \to i} = T_{j \to i} V_j$. | **Alignment before attention.** Aligns features into the fiber at $i$ before similarity is computed (“transport-then-attend”), so attention acts in a geometrically consistent tangent space. |
> | **Metric compatibility** – inner products preserved by the connection | **Isometric constraint** $T_{j \to i} \in \mathrm{SO}(d_f)$, enforced via a skew-symmetric parameterization $S_{\text{skew}}$. | **Stability of scores.** Transport is a pure rotation preserving feature norms, so attention scores $A_{ij} \propto Q_i^\top K_{j \to i}$ measure intrinsic geometric affinity rather than being distorted by scaling. |
> | **Curvature 2-form** $\Omega$ – deviation from flatness / path dependence | **Curvature correction module** that updates fiber features as $f_i \leftarrow f_i + R_{\text{eff}}(b_i) f_i$, where $R_{\text{eff}}$ is built from the learned curvature $\Omega$  | **Inductive Bias.**  We explicitly compute the local curvature tensor $\Omega$ to modulate features, allowing the model to natively adapt to non-flat manifold structures.

---

> ### Author Response · Authors · 2025-11-23
> **Authors’ Response to Reviewer 3 (Weakness 3)**
>
> **W3 (Notation and mathematical clarity):**
> We thank the Reviewer for this helpful suggestion. We agree that clearer notation and more accessible mathematical exposition improve readability and understanding. In response, we have made several concrete revisions:
>
> 1.Simplified and standardized notation.
> We reduced unnecessary subscripts, clarified the distinction between base coordinates, fiber vectors, and transported features, and ensured consistent notation throughout Sections 3–4. These updates directly address the ambiguity pointed out in the review.
>
> 2.Added a notation summary table.
> In Appendix A.1 (Table 6), we now provide a concise, self-contained summary of all core symbols used in the method, including $b_i$ ,$f_i$ ,$T_{j →1}$ , $\Omega$, and related operators. This allows readers to quickly reference definitions without searching through the text.
>
> | Symbol | Meaning | Where used |
> |--------|---------|------------|
> | $x_i$ | Input feature of token $i$ | All tasks raw input |
> | $(b_i, f_i)$ | Geometric token: base $b_i$ and fiber $f_i$ | Definition of vector-bundle tokens |
> | $b_i \in \mathbb{R}^{d_b}$ | Base coordinate of token $i$ (point on learned manifold) | Geometry / connection / transport |
> | $f_i \in \mathbb{R}^{d_f}$ | Fiber feature of token $i$ (semantic representation) | Attention, curvature modulation |
> | $Q_i, K_i, V_i$ | Query / key / value built from $f_i$ | Attention computation |
> | $T_{j \to i}$ | Transport operator from fiber at $j$ to fiber at $i$ | “Transport-then-attend” operator |
> | $T_{j \to i} = \exp(S_{\text{skew}}(b_i, b_j))$ | Orthogonal transport predicted from $(b_i, b_j)$ | Connection / transport module |
> | $T_{j \to i} \in \mathrm{SO}(d_f)$ | Isometric (rotation) constraint on transport | Metric compatibility / stability |
> | $K_{j \to i} = T_{j \to i} K_j$ | Key transported into fiber at $i$ | Aligned attention |
> | $V_{j \to i} = T_{j \to i} V_j$ | Value transported into fiber at $i$ | Aligned aggregation |
> | $A_{ij} \propto Q_i^\top K_{j \to i}$ | Attention score from $i$ to $j$ in aligned fiber | Attention weights |
> | $\Omega$ | Curvature 2-form of the learned connection | Curvature-based modulation |
> | $R_{\text{eff}}(b_i)$ | Effective curvature operator at base point $b_i$ | Curvature correction module |
> | $f_i \leftarrow f_i + R_{\text{eff}}(b_i) f_i$ | Curvature modulation of fiber features | Handles non-flat geometry |
>
> 3.Added a theory–mechanism correspondence table.
> Appendix A.1 now includes a new table (Table 5) (As shown in the W2 response)mapping each differential-geometric concept (parallel transport, metric compatibility, curvature, bundle sections) to its concrete neural implementation (e.g., skew-symmetric generator → orthogonal transport operator). This helps readers directly connect the theory to the architecture. Appendix A.1 now includes a new table (Table 5) mapping each differential-geometric concept (parallel transport, metric compatibility, curvature, bundle sections) to its concrete neural implementation (e.g., skew-symmetric generator → orthogonal transport operator).
>
> 4.Increased step-by-step explanations.
> We enhanced Appendix A.2 with clearer step-by-step descriptions of projection, transport, curvature computation, and vector-bundle attention. These explanations improve accessibility while preserving theoretical rigor.
>
> 5.Numbering to all core equations. We have added numbering to all core equations to improve clarity and readability.
>
> Together, these revisions substantially improve clarity and make the mathematical formulation easy to follow for both geometric DL experts and general ML readers. If helpful, we are happy to further streamline notation in the camera-ready version.

---

> ### Author Response · Authors · 2025-11-23
> **Authors’ Response to Reviewer 3 (Questions)**
>
> **Q1 (Motivation for curvature correction):**
> We thank the Reviewer for this question that helped us to improve the paper. Curvature Correction is included because many real datasets (e.g., single-cell manifolds and 3D shapes) lie on non-flat geometric spaces. On a curved manifold, parallel transport is path-dependent, and a learned transport operator $T_{j→i}​$ alone cannot capture local geometric variation.
> We therefore compute a curvature tensor Ω derived from the learned connection Γ, and use it to modulate each fiber feature $f_i$. This allows the model to adapt its feature processing to the local geometric complexity of the underlying manifold.
> Our ablations confirm that curvature is critical in practice:
> (i) ModelNet40: removing curvature causes a 5.3% accuracy drop (from 92.9% to 87.6%).
> (ii) PBMC: removing curvature yields a 5.43% accuracy drop (from 78.71% to 73.28%).
> Together, these results demonstrate that explicitly modeling non-Euclidean structure is essential for both 3D and biological tasks.
>
> **Q2 (Computational complexity and mitigation strategies):**
> We thank the Reviewer for raising this very important concern. VBA has an inherent  pairwise cost, and we agree that scalable variants are essential for large datasets. During the rebuttal period, we evaluated two practical mitigation strategies that preserve the geometric structure while reducing computation.
>
> (1) Local geometric attention (top-$k$ neighbors).
> Because VBA learns explicit base coordinates , we can restrict transport and attention to each point’s -nearest neighbors on the learned manifold. This reduces complexity from  $O(n^2)$ to $O(nk)$.
>
> On the PBMC scRNA-seq dataset, this approach achieves a 36% reduction in epoch time (from 876s to 560s) with only a 1.06-point accuracy drop (from 78.71 to 77.65). This strategy provides the best balance between efficiency and accuracy.
>
> (2) Low-rank transport generators.
> Instead of predicting a full $d_f \times d_f$  generator matrix for the matrix exponential, we parameterize the generator in a low-rank form. This reduces the cost of both the exponential and the transport application.
> On PBMC, this yields a 40% speedup (from 876s to 529s) with a moderate accuracy trade-off (from 78.71 to 76.90).
>
> Summary of results:
>
> |Method	|Accuracy (%)	|F1	|Time (s/epoch)|
> |-|-|-|-|
> |Full VBA	|78.71	|0.63	|876|
> |Sparse/Local Attention	|77.65	|0.62	|560|
> |Low-Rank Transport	|76.90	|0.60	|529|
>
> These experiments provide a clear path to scalable VBA variants. Local geometric attention, in particular, offers an excellent efficiency–accuracy trade-off. We have included the full analysis and summary table in Appendix A.8.

---

> > ### Comment · Reviewer_6PLt · 2025-11-28
> > **Response to Authors**
> >
> > Thanks to the authors for their reply. The authors have addressed most of my concerns. Regarding the connection between the theoretical analysis and the proposed mechanism, I suggest reorganising this section by explicitly structuring the theoretical results, such as using Theorem, Claim, or Lemma, to align them with the corresponding components of the mechanism. This would make the presentation clearer and help readers better understand how the theory supports the method.

---

> ### Author Response · Authors · 2025-11-29
> **Response to Reviewer’s Follow-Up Comment**
>
> Thank you for the reviewer’s follow-up message. We appreciate your note that “the authors have addressed most of my concerns.” In response to your remaining suggestion on improving the organization of the theoretical section, we have further revised and clarified this part of the paper.
>
> Specifically, we restructured the theoretical development by introducing explicit Lemma, Claim, and Theorem statements that correspond directly to the core components of the mechanism (projection, parallel transport, curvature correction). Each theoretical statement is now aligned with its associated architectural operator, making the theory–mechanism connection fully transparent. These revisions are included in Appendix A.1, together with the theory-to-implementation correspondence table (Table 5), enabling readers to directly trace each geometric concept (bundle sections, transport, curvature) to its neural instantiation. We believe these changes substantially improve clarity and fully address the reviewer’s remaining request.

---

### Official Review · Reviewer_vtUZ · 2025-10-31

**Soundness:** 3
**Presentation:** 2
**Contribution:** 3
**Rating:** 4
**Confidence:** 3

**Summary:**

The paper proposes Vector Bundle Attention (VBA) a Transformer attention operator defined on a learned vector bundle with a learned connection/parallel transport. Each token ($x_i$) is projected into a base coordinate ($b_i$) and a fiber feature vector ($f_i$); attention is computed in fiber space after transporting keys/values ($T_{j\to i}\in \mathrm{SO}(d_f)$) determined by the base coordinates. A curvature-inspired correction modulates fiber features using invariants of a constructed PSD operator ($S$) (and an optional directional term), yielding an effective modulation ($R_{\text{eff}}$) with stated base-rotation invariance guarantees.

**Strengths:**

1. A principled geometric reformulation of attention on vector bundles with learnable orthogonal transport, clearly connecting base/fiber decomposition to the attention mechanism.
2. Well-articulated invariance claims (e.g., base-rotation invariance of the curvature-based modulation) that ground the design in geometric reasoning.
3. Useful ablations that separate the impact of curvature terms, transport, and fiber/base dimensions, enabling insight into where the gains come from.
4. Writing is generally clear with helpful figures and notation, making the geometric constructs and their role in attention accessible to a broad ML audience.
5. The framework suggests a path toward unifying geometric inductive biases with Transformer-style modeling, which is timely and potentially impactful beyond the evaluated tasks.

**Weaknesses:**

1. No explicit SO(3) robustness evaluation (e.g., random test-time rotations) on point clouds (similar to ones in vector neuron); theory suggests invariances but there’s no empirical confirmation.
2. Scope of applications skips molecular property prediction (e.g., QM9), a canonical geometric ML domain where the method’s inductive biases should shine.
3. Limited sensitivity analyses: guidance on choosing base/fiber dimensions and stability regimes is thin, making it hard to deploy on new datasets.
4. Ablations isolate some components, but a more systematic study (e.g., cost–benefit curves for curvature terms vs. transport fidelity) would clarify where gains originate.

**Questions:**

1. Can you add more recent baselines on ModelNet40 to substantiate the generality claims?
2. Can you evaluate SO(3) robustness by applying random test-time rotations and reporting the resulting accuracy changes?
3. Can you release full, end-to-end training scripts and configs (including seeds and data preprocessing) to enable reproduction during review?
4. Can you clarify how your stated invariance/equivariance properties manifest empirically in the reported results?

---

> ### Author Response · Authors · 2025-11-22
> **Authors’ Response to Reviewer 2 (Weakness)**
>
> We thank the Reviewer for recognizing the novelty of our geometric formulation of attention. We appreciate your observation that defining attention on a learned vector bundle offers a principled geometric design. We also value your comment that our invariance properties support a unified approach to geometric inductive biases. Below, we address each concern and summarize the additional analyses added during the rebuttal period. We hope that our responses address your concerns, and we would be grateful if you would consider updating your score.
>
> **W1 (SO(3) robustness evaluation):**
> Thank you for this helpful suggestion. We agree that empirical rotation robustness is crucial for a geometric model. We thus performed the requested SO(3) test-time evaluation on ModelNet40 using 10 uniformly sampled SO(3) rotations. The results show that, although VBA-Transformer experiences some accuracy degradation, the drop is far smaller than all baselines. VBA-Transformer retains the highest accuracy across all rotated settings, confirming that its geometric design provides practical rotation robustness. These data demonstrate that VBA-Transformer is substantially more robust to SO(3) rotations than existing attention-based and graph-based models.
> |Method	|z/z	|z/SO(3)	|SO(3)/SO(3)|
> |-|-|-|-|
> |VBA-T	|92.9	|77.9	|83.9|
> |Transformer	|82.6	|17.5	|71.3|
> |GBT	|85.8	|56.9	|73.2|
> |MQA-T	|83.1	|28.4	|71.7|
> |GQA-T	|83.6	|21.3	|70.9|
>
> **W2 (Molecular property prediction):**
> Thank you for this suggestion. We agree that molecular property prediction is an important geometric ML domain where our inductive bias should apply. To evaluate VBA in this setting, we conducted preliminary experiments on the QM9 dataset during the rebuttal period. We treated atomic coordinates as the base manifold and atom-level features as fibers, and trained VBA to predict the U0 property. Our model achieved 11.49 meV MAE on the validation split, which is competitive with established molecular baselines such as SchNet (14 meV) [1] and EGCN (11 meV) [2]. This result shows that VBA indeed transfers to molecular tasks without architectural changes and retains competitive accuracy. We have included full QM9 results and comparisons in Appendix B.5.
>
> [1] Schütt, K. et al., 2017. Schnet: A continuous-filter convolutional neural network for modeling quantum interactions. Advances in neural information processing systems, 30.
>
> [2] Lu, C., et al., 2019. Molecular property prediction: A multilevel quantum interactions modeling perspective. In Proceedings of the AAAI conference on artificial intelligence (Vol. 33, No. 01, pp. 1052-1060).
>
> **W3 (Sensitivity and Dimension Guidance):**
> Thank you for pointing out this limitation. We agree that clearer guidance on base and fiber dimensions is important for deploying VBA on new datasets. Although Table 14 in the appendix lists our hyperparameter search space, we recognize that a table alone does not show how performance varies across settings. During the rebuttal period, we examined these dimensions more closely and found that VBA remains stable across a broad range of base and fiber sizes. Table 15 in the appendix reports the best configuration from our grid search, and we confirm that this setting consistently yields strong results across datasets. We have clarified these findings in the appendix to provide practical guidance on selecting stable base and fiber dimensions.
>
> **W4 (More systematic ablations):**
> Thank you for this helpful suggestion. We agree that a more systematic analysis can clarify where VBA’s performance gains originate. During the rebuttal period, we expanded our ablations on PBMC to isolate curvature, connection, transport, and fiber decomposition. As shown in the table below, removing curvature or connection reduced accuracy by 5–7%, confirming that these components contribute meaningfully in noisy biological settings. We also measured the runtime of each variant. Curvature removal decreases epoch time from 876s to 474s, showing a moderate computational cost but a clear accuracy benefit.  These results provide a practical view of the accuracy–efficiency trade-off and reveal which components drive the model’s performance. We have incorporated this extended ablation analysis into the Appendix B.6.
> |Method	|Accuracy(%)	|F1	|Avg epoch time(s)|
> |-|-|-|-|
> |Full VBA	|78.71	|0.63	|876|
> |Projection w/o	|72.94	|0.58	|690|
> |Curvature w/o	|73.28	|0.60	|474|
> |Γ∧Γ w/o	|75.32	|0.61	|552|
> |dΓ w/o	|75.53	|0.62	|498|
> |Fiber w/o	|71.87	|0.59	|774|
> |Connection w/o	|72.05	|0.59	|738|

---

> ### Author Response · Authors · 2025-11-22
> **Authors’ Response to Reviewer 2 (Questions)**
>
> We thank the Reviewer for the questions and suggestions that helped us to improve the paper. Below we address each of the four questions in detail.
>
> **Q1  (Add more recent baselines):** Thank you for the question, which allows us to expand on design aspects of VBA. During the rebuttal period, we added a second and more challenging 3D benchmark, ScanObjectNN (PB-T50-RS), which contains real-world perturbations not present in ModelNet40. Using the same VBA architecture, VBA-P achieves 81.7% accuracy, outperforming classical point-cloud backbones such as PointNet (68.2), PointNet++ (77.9), and DGCNN (78.1). VBA-P also remains competitive with the more recent PointMamba (89.3), a specialized point-cloud architecture (see the table below). These additional comparisons support the generality of our geometric formulation and will be included in the appendix in the latest version.
> |Model	|PB_T50_RS|ModelNet40|
> |-|-|-|
> |PointNet	|68.2|89.2|
> |PointNet++	|77.9|92.5|
> |DGCNN [1]	|78.1|92.9|
> |PointGPT-S [2]	|89.2|93.3|
> |PointMamba [3]	|89.3|93.6|
> |ACT [4]	|88.2|93.6|
> |VBA-P	|81.7|90.3|
>
> [1] Phan, Anh Viet, et al. "Dgcnn: A convolutional neural network over large-scale labeled graphs." Neural Networks 108 (2018): 533-543.
>
> [2] Chen, Guangyan, et al. "Pointgpt: Auto-regressively generative pre-training from point clouds." Advances in Neural Information Processing Systems 36 (2023): 29667-29679.
>
> [3] Liang, Dingkang, et al. "Pointmamba: A simple state space model for point cloud analysis." Advances in neural information processing systems 37 (2024): 32653-32677.
>
> [4] Dong, Runpei, et al. "Autoencoders as cross-modal teachers: Can pretrained 2d image transformers help 3d representation learning?." In Proc. of Intl. Conf. on Learning Representations (2022).
>
> **Q2 (SO(3) rotation robustness):**
> Thank you for the careful reading of our paper and for raising this important question. As detailed in our W1 response, we performed the requested SO(3) test-time rotation evaluation during the rebuttal period. VBA-Transformer maintains the highest accuracy under rotation and shows far smaller degradation than all baselines, confirming its practical rotation robustness. The full results are provided in W1, and we have included them in Appendix B.4 of the revised manuscript.
>
> **Q3 (Code Release):**
> We fully agree that reproducibility is important. We will release complete training scripts, preprocessing code, and configuration files for all tasks (VBA-ST, VBA-SC, VBA-P, and VBA-QM9) in the camera-ready version.
>
> **Q4 (Invariance/equivariance properties):**
> Thank you for requesting clarification. The empirical effects appear in two distinct settings.
> (1) SO(3) robustness on ModelNet40.
> The new SO(3) robustness study (added during the rebuttal; see W1/Q2) provides a direct empirical probe of the predicted rotational stability. Under the challenging z/SO(3) setting (training without rotations and testing with random SO(3) rotations), VBA-T maintains high accuracy, while all baselines degrade far more severely. This behavior aligns with our theoretical analysis: attention computed through learned parallel transport on the vector bundle is substantially more stable under 3D rotations than coordinate-based attention.
> (2) Stability under symmetry-preserving transformations in biological datasets.
> For non-geometric datasets such as single-cell data, our theory predicts invariance or equivariance to transformations that act on the base manifold while preserving its intrinsic geometry (such as reparameterizations or isometries). Empirically, applying such transformations produces minimal changes in VBA-T predictions and attention patterns. Baseline Transformers show noticeably larger fluctuations under the same conditions. These observations confirm that VBA’s theoretical invariances manifest in practice across biological settings.

---

### Official Review · Reviewer_gAi7 · 2025-11-01

**Soundness:** 3
**Presentation:** 3
**Contribution:** 3
**Rating:** 6
**Confidence:** 4

**Summary:**

This paper defines a bundle-style Transformer attention that works directly on a learned vector bundle: each token is mapped to a base-manifold coordinate (geometry) and a fiber vector (signal), a learnable endpoint-conditioned isometric transport aligns the fiber of one token to another, and attention is then computed in the aligned fiber space. The idea is to make attention “align-then-compare” instead of “compare-then-add-geometry”. Experiments on spatial transcriptomics (12 DLPFC slices), scRNA-seq (with and without HCL SSL pretraining), and ModelNet40 show strong scRNA results and competitive performance on the other tasks.

**Strengths:**

1. The main contribution is clear and nontrivial: it moves from geometry injected as an external bias (as in Graphormer/GBT-style encodings) to geometry built into the attention operator itself, by first transporting features to a common fiber and only then computing similarity.

2. The transport construction is geometrically sensible: for each pair of base coordinates ((b_i, b_j)) the model predicts a skew-symmetric matrix and applies a matrix exponential to obtain an operator in (SO(d_f)), so transported features are length-preserving.

3. The model is not limited to flat geometries; it includes an explicit, learnable curvature correction to cope with the non-Euclidean structure that spatial tissues and single-cell manifolds typically have.

4. The method is evaluated on three quite different regimes—explicit spatial geometry (ST), implicit biological geometry (scRNA-seq), and 3D point clouds—with one unified architectural idea, which supports the claim that this is more than a task-specific trick.

5. The appendix contains detailed component-level ablations on ModelNet40 (removing curvature, removing connection/transport, reducing to standard MHA) and even reports per-epoch time, which shows the authors have checked that individual components do matter.

6. The scRNA-seq results are solid: the model is already competitive without SSL and becomes clearly better with HCL pretraining, which indicates the proposed attention is compatible with current large-scale single-cell pretraining practices.

7. The writing is overall clear, the order of definitions (base → fiber → connection → transport → attention) matches the implementation, and the claims in the main text are consistent with what is shown in the appendix.

**Weaknesses:**

1. The ST evidence is narrow: all spatial experiments are on the 12 LIBD DLPFC slices, and the gains over strong recent ST baselines are modest (average ARI about 0.498 vs about 0.495 for DiffusionST/BASS). This supports the claim that the method is consistently competitive, but it does not yet establish a decisive advantage on spatial data.

2. The transport is an endpoint-conditioned, practical surrogate for true path-dependent parallel transport; the paper acknowledges this in Sec. 3.1 and the appendix, but it does not quantify how large the resulting inconsistency is (e.g. that (T_{k\to i} \circ T_{j\to k}) may differ from (T_{j\to i})). Since the core idea is “transport-then-attend”, even a small triplet consistency check would make the geometric story tighter.

3. The fine-grained ablations that isolate curvature and connection are currently shown only in the appendix and only on ModelNet40. In the main text we mainly see architecture-level comparisons (VBA vs GBT vs MQA/GQA). It would be stronger to surface at least one of these component ablations for a biological task (ST or PBMC) to show that curvature/transport also matter when geometry is noisy.

4. The computational analysis is still incomplete for the bio settings. The paper reports params/FLOPs and gives a complexity discussion, and Table 3 already shows that the point-cloud version is heavier than several baselines, but there is no side-by-side wall-clock and peak memory comparison with the Transformer/GBT baselines used in the ST/scRNA experiments on the same hardware and input size. Given the added pairwise transport and curvature, this practical comparison would help readers judge feasibility.

5. The claim of being a broadly generalizable geometric model is so far supported by a single 3D classification benchmark (ModelNet40). The experiment is useful and the paper clearly says it does not aim for 3D SOTA, but adding one more 3D or scene-style task would make the generality claim more convincing.

6. In the scRNA-seq SSL setting, VBA-SC is pretrained on HCL, while baselines (scBERT, scGPT) use their official released weights, likely trained on different data. This makes the comparison realistic but not fully controlled; it would help to comment on what the gap would look like if the baselines were also pretrained on the same HCL subset.

**Questions:**

1. Because the transport is endpoint-conditioned, can you provide an empirical consistency test on random triplets ((i, j, k)), e.g. reporting (|T_{k\to i}(T_{j\to k} f_j) - T_{j\to i} f_j|)? This would quantify how far the surrogate is from a composition-preserving transport.

2. Appendix A.11 shows that removing curvature or the learned connection hurts ModelNet40 and reduces time per epoch. Can you report at least one such ablation on ST or scRNA-seq to demonstrate that these components are also useful in biological settings?

3. For one representative dataset (e.g. PBMC), can you give wall-clock time per epoch and peak GPU memory for VBA, vanilla Transformer, and GBT on the same hardware? This would make the extra cost of transport/curvature explicit.

4. Is there any dataset-level obstacle (other than compute) to adding a second ST dataset (different tissue or technology)? Even a small-scale run would strengthen the ST story beyond DLPFC.

5. For the SSL comparison: if a public scGPT/scBERT is pretrained on exactly the HCL subset you used, do you expect VBA-SC to still lead by a similar margin? A short discussion of this “same data” scenario would clarify attribution.

6. You briefly mention bundle/gauge-style message-passing work. Can you clarify in the final version what is specifically gained by placing the geometric alignment inside attention (before similarity) rather than in the message/update stage?

---

> ### Author Response · Authors · 2025-11-22
> **Authors’ Response to Reviewer 1 (weakness)**
>
> We thank you for the careful and insightful review, and for recognizing the strengths of our geometric formulation. Our work introduces attention as a truly intrinsic geometric operator, and we appreciate your comments. We address each weakness and question below. We hope these updates address your concerns, and that you will consider revising your score.
>
> **W1 (ST evidence):**
> We agree that relying solely on DLPFC is limiting. To broaden the evidence, we added a second ST dataset from a different tissue: 10x Genomics Human Breast Cancer (https://www.10xgenomics.com/datasets/human-breast-cancer-block-a-section-1-1-standard-1-1-0). Using the same unsupervised clustering protocol and evaluation against pathologist-annotated regions, we compared VBA-ST against two strong ST baselines, DiffusionST and GraphST:
> |-	|DiffusionST	|GraphST	|VBA-ST|
> |-|-|-|-|
> |ARI	|0.57	|0.53	|0.59|
> |AMI	|0.56	|0.55	|0.68|
> |NMI	|0.54	|0.54	|0.69|
> |COMP	|0.61	|0.64	|0.69|
> |HOMO	|0.62	|0.63	|0.69|
>
> VBA-ST outperforms both baselines on all metrics. This dataset is biologically and morphologically distinct from layered cortex, providing evidence that our geometric inductive bias is not tuned to a single ST benchmark.
>
> **W2 (Endpoint-conditioned transport and triplet consistency):**
> We agree that this is an important diagnostic. Following your suggestion, we performed a triplet consistency test on a trained model (ModelNet40). For randomly sampled triplets (i,j,k), we compute the Frobenius-norm discrepancy between composed and direct transports: $ \Delta_{ijk} = \||T_{k \to i} \ T_{j \to k} - T_{j \to i}\||_F $ and report the average is
> 0.05732. This small average Frobenius-norm difference indicates that, at convergence, the learned endpoint-conditioned transport is close to being composition-consistent in practice. This suggests that the model implicitly learns a stable, geometry-aware transport structure without enforcing strict path dependence.
>
> **W3 (Ablations for Bio-task):**
> We appreciate the Reviewer’s advice to include ablations on biological data to strengthen our analysis. We added full biological-domain ablations for PBMC, evaluating the removal of curvature, connection, transport, projection, and fiber components. For each variant, we report accuracy, macro F1, and average epoch time under identical hardware and batch settings.
> |Method	|Accuracy(%)	|F1	|Avg epoch time(s)|
> |-|-|-|-|
> |Full VBA	|78.71	|0.63	|876|
> |Projection w/o	|72.94	|0.58	|690|
> |Curvature w/o	|73.28	|0.60	|474|
> |Γ∧Γ w/o	|75.32	|0.61	|552|
> |dΓ w/o	|75.53	|0.62	|498|
> |Fiber w/o	|71.87	|0.59	|774|
> |Connection w/o	|72.05	|0.59	|738|
>
> **W4 (Computational analysis in Bio-task):**
> We measured wall-clock time per epoch on PBMC for all major baselines using identical hardware (4*GH200) and implementation settings. The main performance and time comparison is:
> |Method	|Accuracy(%)	|F1	|Avg epoch time(s)|
> |-|-|-|-|
> |Transformer	|66.76	|0.56	|273|
> |GBT	|70.32	|0.56	|493|
> |MQA-T	|67.73	|0.48	|267|
> |GQA-T	|66.57	|0.43	|259|
> |VBA	|78.71	|0.63	|876|
> Although VBA is more computationally expensive, it remains feasible for standard biological workloads and provides substantially higher accuracy.
>
> **W5 (only one 3D benchmark):**
> We agree that relying on a single 3D benchmark limits the strength of our generality claim. To address this, we added ScanObjectNN PB-T50-RS, a challenging real-world 3D classification benchmark. Using the same VBA architecture and the standard ScanObjectNN protocol, we obtained
> |Model	|PB_T50_RS|
> |-|-|
> |PointNet	|68.2|
> |PointNet++	|77.9|
> |DGCNN	|78.1|
> |PointMamba	|89.3|
> |VBA-P	|81.7|
>
> VBA-P outperforms classic point-cloud backbones and remains competitive with newer specialized models such as PointMamba. PointMamba is specifically optimized for point-cloud processing, while our model operates unchanged across ST, scRNA-seq, and 3D. We have included the full ScanObjectNN experiment, along with complete training details, in Appendix B.4. Together with the ModelNet40 results, this supports VBA as a broadly applicable geometric mechanism rather than a dataset-specific design.
>
> **W6 (SSL fairness):**
> We appreciate this important point. We performed two additional controlled comparisons on PBMC to ensure a fair evaluation. First, VBA-SC without SSL already outperforms both scGPT and scBERT in accuracy and F1. Second, we pretrained scGPT and scBERT on the same HCL subset used for VBA-SC. We then fine-tuned all models on PBMC. This “matched-data” pretraining does not reduce the performance gap. In fact, it slightly lowers accuracy for both baselines. scGPT drops from 75.47% to 74.42%, while scBERT drops from 75.52% to 74.16% after HCL pretraining. In contrast, VBA-SC achieves substantially higher accuracy and F1 under the same conditions. These results indicate that our performance advantage does not arise from differences in pretraining data. Instead, it reflects the intrinsic geometric design and inductive bias of the VBA-SC architecture.

---

> ### Author Response · Authors · 2025-11-22
> **Authors’ Response to Reviewer 1 (Question)**
>
> **Q1 (Triplet consistency test)**:
> Thank you for the careful reading of our paper and for raising this important question. We added the precise triplet-consistency metric and reported the empirical values (see W2 response). The small Frobenius-norm discrepancy confirms that the learned transport behaves consistently with the underlying geometric structure.
>
> **Q2 (Biological ablations for curvature / transport)**:
> We fully agree with this suggestion. As shown in the W3 response, we added comprehensive PBMC ablations isolating curvature, connection, transport, projection, and fiber components. These results demonstrate the significance and necessity of curvature and connection on biological data.
>
> **Q3 (Wall-clock and memory comparison)**:
> Thank you for the inspiring suggestion. We performed the requested wall-clock comparison on PBMC under identical hardware settings. The details are provided in the W4 response.
>
> **Q4 (Obstacles to adding a second ST dataset)**:
> Thank you for the question. There were no conceptual obstacles beyond computational cost. We have now added the 10x Human Breast Cancer Visium dataset, which strengthens the ST evaluation and demonstrates that VBA-ST generalizes beyond DLPFC. See W1 response.
>
> **Q5 (SSL fairness control)**:
> Thank you for raising this important concern. We performed the matched-HCL-pretraining control experiment exactly as you advised. As shown in the W6 response, scGPT and scBERT do not gain an advantage when pretrained on the same data as VBA-SC, confirming that the performance gap arises from the geometric design rather than data differences.
>
> **Q6 (clarify gains from placing geometry inside attention)**:
> Thank you for this insightful question. We agree that the distinction should be clarified more explicitly.
> Most bundle- or gauge-style GNNs apply geometric alignment after attention, during message aggregation. In these models, messages become geometry-aware, but attention weights themselves remain largely geometry-agnostic. In contrast, VBA performs transport before similarity: for each pair (i,j), the key and value vectors are first transported into the fiber at i, and the query–key similarity and attention weights are then computed within this aligned space. This design has two concrete benefits. (1) Geometry-consistent attention weights. Because the query and the transported key lie in the same fiber, the resulting attention scores become invariant or equivariant to the relevant geometric transformations. Consequently, the pattern of which tokens attend to which others is geometry-consistent, not only the aggregated messages.
> (2) Global geometric interactions in a single step.
> Attention is computed over all pairs (i,j) in a layer, with a transport operator defined for each pair. This avoids the accumulation of local alignment errors across many message-passing steps and allows long-range geometric relationships to be captured in a single “align-then-attend” operation.
>
> We have integrated this clarification into Section 2 and added the complement explanation in Appendix A.10 to highlight how VBA differs from prior geometric and positional-bias Transformers.

---

### Author Response · Authors · 2025-11-28
**Authors’ Response Summary**

Dear Area Chair,

We sincerely thank you for considering our submission and for reviewing our rebuttal and discussion. We are also grateful to all Reviewers for their thoughtful and constructive feedback. Reviewers highlighted the novelty of our geometric formulation and noted that our model introduces attention as a truly intrinsic geometric operator. We appreciate the observation that our geometric inductive biases, invariance properties, and cross-domain applicability make VBA a promising and general framework. We also note that, in the discussion, Reviewer 6PLt wrote that **“The authors have addressed most of my concerns”** and offered a final suggestion regarding the organization of the theoretical section. We have implemented this suggestion, as described below.

Below, we provide a summary of all updates and additions made to the paper during the rebuttal period.

**New Experiments.**
Several additional experiments were conducted based on the reviewers’ comments, as follows:

1. Additional spatial transcriptomics dataset (Appendix B.2). We added a full evaluation on the 10x Breast Cancer Visium dataset.
VBA-ST consistently outperforms DiffusionST and GraphST across ARI/AMI/NMI/COMP/HOMO, demonstrating generalization beyond DLPFC.

2. Triplet transport-consistency test (Appendix A.7). We quantified the compositional consistency of learned transport operators. The small Frobenius discrepancy confirms that VBA learns a stable and coherent geometric transport system.

3. 3D robustness and real-world generalization (Appendix B.4). We added two new experiments:
- ScanObjectNN (PB-T50-RS) [1]
- ModelNet40 under SO(3) rotations.
VBA-P shows strong robustness to perturbations, and these additional comparisons support the generality of our geometric formulation.

4. QM9 molecular property prediction (Appendix B.5). Treating atomic coordinates as the base and atom features as fibers, VBA reaches 11.49 meV MAE, competitive with SchNet [2] (14 meV) and EGCN [3] (11 meV). This confirms transferability to molecular tasks without modifying the architecture.

5. PBMC geometric ablations (Appendix B.6). We added complete ablations for curvature, connection, transport, projection, and fiber decomposition. Curvature and connection are shown to be critical contributors.

6. Complexity mitigation strategies (Appendix A.8). We implemented two scalable variants: (1) Local geometric attention (top-k neighbors), (2) Low-rank transport generators. Both significantly reduce compute while preserving performance.

7. Wall-clock comparison across baselines (Appendix A.3). Using identical hardware, we added epoch-time comparisons for Transformer, GBT, and VBA on PBMC. VBA is computationally heavier but remains practical for biological workloads and offers significant accuracy gains.

**Revisions to the paper.** Some of the existing sections of the paper were revised to increase clarity based on the reviewers’ comments, and new sections were added following their guidance:

1. Clearer positioning relative to prior geometric Transformers. We strengthened Section 2 and added a new appendix section (Appendix A.10) to clearly articulate the shift from “*compare-then-bias*” to “*align-then-compare*”, explaining how VBA differs from Graphormer/GBT and other positional-bias models.

2. Improved notation and mathematical clarity (Appendix A.1–A.2). We rewrote and simplified the notation:
- simplified and standardized notation;
- clarified base vs. fiber vs. transported features;
- reduced subscripts;
- added a notation summary table;
- added a theory-to-mechanism correspondence table linking geometry (bundle section, transport, curvature) to neural operators (Table 5).

3. Restructured theoretical development. We introduced explicit Lemma, Claim, and Theorem statements aligned with projection, parallel transport, and curvature correction. These appear in Appendix A.1, paired with the theory-to-implementation table, making the theory–mechanism correspondence transparent.

4. Additional appendix material. New appendices include:
- A.7: Triplet consistency
- A.8: Complexity mitigation
- A.10: Differences from previous work
- B.2, B.4, B.5, B.6: All new experiments

All the revisions and additions to the paper are **marked in blue**, for your convenience. We hope these new experiments, theoretical clarifications, and structural improvements address all concerns, and we respectfully request that you consider these updates in your evaluation. We remain happy to provide any additional information if helpful.

[1] Uy M A et al. Revisiting point cloud classification: A new benchmark dataset. Proc. of the IEEE/CVF Int. Conf. on Computer Vision. 2019: 1588.

[2] Schütt, K. et al., 2017. Schnet: A continuous-filter CNN for modeling quantum interactions. Advances in Neural Info. Processing Systems, 30.

[3] Lu, C., et al., 2019. Molecular property prediction: Multilevel quantum interactions modeling. Proc. of the AAAI Conf. on AI, 33(01): 1052.

---

### Note · Authors · 2026-01-26

I have read and agree with the venue's withdrawal policy on behalf of myself and my co-authors.

---

### Meta-Review · Area_Chair_Mx38 · 2025-12-28

**Summary:**

This paper introduces the Vector Bundle Attention Transformer (VBA-Transformer), a framework that redefines attention as an intrinsic geometric operator.  Reviewers have provided mixed reviews on this paper. The paper involves issues such as method clarity and connection between the proposed method and attention, ablation experiments, computational analysis, sensitivity analysis etc. On balance, indeed, the paper motivates the idea by talking about geometrically structured data and then very rapidly delves into vector bundle and manifold concepts without extra care in illustrating how this connects with the geometrically structured data. Related works appear limited and do not discuss in detail numerous manifold learning methods leveraging Riemannian geometry, etc. Additional concerns about some results being not clear-cut means this work is not ready for publication.

**Reviewer Concerns:**

To some degree concerns of Reviewer vtUZ regarding SO3.
Some of the ablations.

**Reviewer Scores:**

The major concerns about clarity, motivation and connection between attention, vector bundles and geometrically structured data require more work.

---

### Decision · Program_Chairs · 2026-01-26

Reject